# Observation of Bloch oscillations dominated by effective anyonic particle statistics

Weixuan Zhang [1,3], Hao Yuan[1,3], Haiteng Wang[1], Fengxiao Di[1], Na Sun[1], Xingen Zheng [1], Houjun Sun [2] &
Xiangdong Zhang [1✉]

Bloch oscillations are exotic phenomena describing the periodic motion of a wave packet subjected to an external force in a lattice, where a system possessing single or multiple particles could exhibit distinct oscillation behaviors. In particular, it has been pointed out that quantum statistics could dramatically affect the Bloch oscillation even in the absence of particle interactions, where the oscillation frequency of two pseudofermions with an anyonic statistical angle of $\pi$ becomes half of that for two bosons. However, these statistically dependent Bloch oscillations have never been observed in experiments until now. Here, we report the experimental simulation of anyonic Bloch oscillations using electric circuits. By mapping the eigenstates of two anyons to the modes of the designed circuit simulators, the Bloch oscillations of two bosons and two pseudofermions are verified by measuring the voltage dynamics. The oscillation period in the two-boson simulator is almost twice of that in the two-pseudofermion simulator, that is consistent with the theoretical prediction. Our proposal provides a flexible platform to investigate and visualize many interesting phenomena related to particle statistics and could have potential applications in the field of the signal control.

[1] Key Laboratory of advanced optoelectronic quantum architecture and measurements of Ministry of Education, Beijing Key Laboratory of Nanophotonics & Ultrafine Optoelectronic Systems, School of Physics, Beijing Institute of Technology, 100081 Beijing, China. [2] Beijing Key Laboratory of Millimeter Wave and Terahertz Techniques, School of Information and Electronics, Beijing Institute of Technology, 100081 Beijing, China. [3] These authors contributed equally: Weixuan Zhang, Hao Yuan. ✉email: zhangxd@bit.edu.cn

Bloch oscillations (BOs) were originally proposed for electrons in crystals, which are characterized as the coherent oscillatory motion of electrons in a periodic potential driven by an external DC electric field[1,2]. After a long-lasting debate about the correctness of BOs[3,4], the effective Hamiltonians leading to BOs and their frequency-domain counterparts, called the Wannier-Stark ladder, were confirmed[5]. The first experimental observation of BOs was based on semiconductor superlattices[6], and a few years later, atoms in an optical potential were also established to demonstrate such a novel effect[7–10]. In fact, BO is a universal wave phenomenon. Hence, it has also been observed in various classical wave systems[11–20], such as coupled optical waveguides[11–19] and acoustic superlattices[20].

On the other hand, in the few-body quantum systems described by the Bose-Hubbard or Fermi-Hubbard model, many investigations have shown that BOs could be significantly modified by strong particle interactions[20–29]. In particular, the frequency doubling of BOs for two strongly correlated Bosons, which is called fractional BOs, is experimentally observed based on a photonic lattice simulator[29], where the two-boson dynamics are directly mapped to the propagation of light fields in the designed two-dimensional waveguide array.

Except for bosons and fermions, anyons are quantum quasiparticles with statistics intermediate between them[30–36]. Anyons play important roles in several areas of modern physics research, such as fractional quantum Hall systems[37–40] and spin liquids[41–43]. Another potential application of non-Abelian anyons is to realize topological quantum computation[44]. Interestingly, previous theoretical work has shown that two noninteracting anyons could exhibit exotic BOs, where the frequency halving of BOs for two pseudofermions exists if the ratio of the applied external force to the hopping rate is less than or equal to 0.5[45]. However, the experimental observation of anyonic BOs is still a great challenge in condensed-matter systems, ultracold quantum gases, and other classical wave systems. In this case, a newly accessible and fully controllable platform should be constructed to simulate anyonic BOs with novel behaviors.

In this work, we demonstrate both theoretically and experimentally that anyonic BOs can be simulated by designed electric circuits. Using the exact mapping of two anyons in the external forcing to modes of designed circuit lattices, the periodic breathing dynamics of voltages are observed by time-domain measurements in both two-boson and two-pseudofermion circuits simulators. In particular, we find that the oscillation frequency in the two-boson simulator is almost twice that in the two-pseudofermion simulator, which is consistent with the theoretical prediction. Our work provides a flexible platform to implement many interesting phenomena depending on particle statistics and could have potential applications in the field of intergraded circuit design and electronic signal control.

## Results

### The theory of simulating anyonic Bloch oscillations by electric circuits. 

Following the theoretical model proposed by Longhi and Valle[45], we start by considering a pair of non-interacting anyons hopping on a one-dimensional (1D) chain subjected to an external force $F$. In this case, the system can be described by the tight-binding lattice model as:

$$H = -J\sum_{l=1}^{N}(a_l^+ a_{l+1} + a_{l+1}^+ a_l) + F\sum_{l=1}^{N} ln_l, \quad (1)$$

where $a_l^+$ ($a_l$) and $n_l = a_l^+ a_l$ are the creation (annihilation) and particle number operators of the anyon at the $l$th lattice site, respectively. $N$ is the number of lattice sites. $J$ is the single-particle hopping rate between adjacent sites. The anyonic creation

and annihilation operators obey the generalized commutation relations as:

$$a_k^+ a_l - a_l^+ a_k e^{i\theta sgn(l-k)} = \delta_{lk}, a_l a_k - a_k a_l e^{i\theta sgn(l-k)} = 0, \quad (2)$$

where $\theta$ is the anyonic statistical angle, and $sgn(x)$ equals $-1$, 0, and 1 for $x < 0$, $x = 0$ and $x > 0$, respectively. It is worth noting that anyons with $\theta = \pi$ are "pseudofermions" as they behave like fermions off-site while being bosons on-site. The two-anyon solution can be expanded in the Fock space as:

$$|\psi> = \frac{1}{\sqrt{2}} \sum_{m,n=1}^{N} c_{mn} a_m^+ a_n^+ |0>, \quad (3)$$

where $|0>$ is the vacuum state and $c_{mn}$ is the probability amplitude with one anyon at site $m$ and the other one at site $n$. Under the restriction of anyonic statistics, the relationship of $c_{mn} = e^{i\theta sgn(n-m)} c_{nm}$ is satisfied. Substituting Eqs. (1) and (3) into the Schrödinger equation $H|\psi> = \varepsilon|\psi>$, we obtain the eigen-equation with respect to $c_{mn}$ as:[45]

$$\varepsilon c_{mn} = -J[e^{i\theta(\delta_{m,n}+\delta_{m+1,n})}c_{m(n-1)} + e^{-i\theta(\delta_{m,n}+\delta_{m-1,n})}c_{m(n+1)} \\ + c_{(m-1)n} + c_{(m+1)n}] + F(m+n)c_{mn}. \quad (4)$$

Similar to many previous works on mapping lower-dimensional few-body systems to higher-dimensional single-body system[29,46–49], We note that Eq. (4) can also be regarded as the eigen-equation describing a single-particle hopping on the 2D lattice with the spatially modulated on-site potential and hopping configuration, as shown in Fig. 1a. In this case, the probability amplitude for the 1D two-anyon model with one anyon at site $m$ and the other at site $n$ is directly mapped to the probability amplitude for the single particle located at site $(m, n)$ of the 2D lattice. The position-dependent on-site potential could simulate the effect of external force. Moreover, the hopping of a single particle along a certain direction in the 2D lattice represents the hopping of one anyon in the 1D lattice. In this case, the behavior of two anyons in the 1D lattice can be effectively simulated by a single particle in the mapped 2D lattice, which inspires the design of classical simulators to study statistic-dependent anyonic physics.

One of the fascinating phenomena dominated by the quantum statistics in Eq. (4) shows that the BO frequency of two pseudofermions ($\theta = \pi$) becomes half of that for two bosons ($\theta = 0$), when the ratio of the external forcing to the hopping rate satisfies $F/J \leq 0.5$[45]. To clarify this effect, the evolution of two-anyon eigen-energies as a function of $\theta$ is displayed in Fig. 1b with $J = 1$ and $F = 0.5$. For a clear illustration. eigen-energies in the range of $(6F, 24F)$ are plotted. Moreover, to avoid the finite-size effect, only eigen-energies with their eigenmodes showing the largest overlap with the center of lattices are kept. As for the case of two bosons, the mapped 2D lattice in Fig. 1a possesses a mirror symmetry with respect to the $m = n$ line, which protects the degeneration of different energy-levels (blue dots for $\theta = 0$), and the degenerated eigen-energies $\varepsilon_{mn} = (m+n)F$ are equally spaced in the form of the Wannier-Stark ladder with $\Delta\varepsilon = F$. By introducing the statistical angle of two anyons, the mirror symmetry is broken, resulting in a splitting of the highly degenerated eigen-spectrum of two bosons. In this case, the eigen-spectra of two anyons are always not equal-spaced and the energy spacing is smaller than the bosonic counterpart. When the statistical angle reaches to $\theta = \pi$, many eigenmodes become nearly degenerated again. Under a relatively strong hopping condition ($F/J < 0.5$), the suitable energy-level coupling between different anyonic bands could make the Wannier-Stark spectrum reappear for pseudofermions. The lower spatial symmetry of the mapped lattice model of pseudofermions compared to that of bosons leads to a smaller energy degeneracy and a denser distribution of eigen-spectrum. In this case, the energy spacing of pseudofermions is $\Delta\varepsilon = F/2$, which

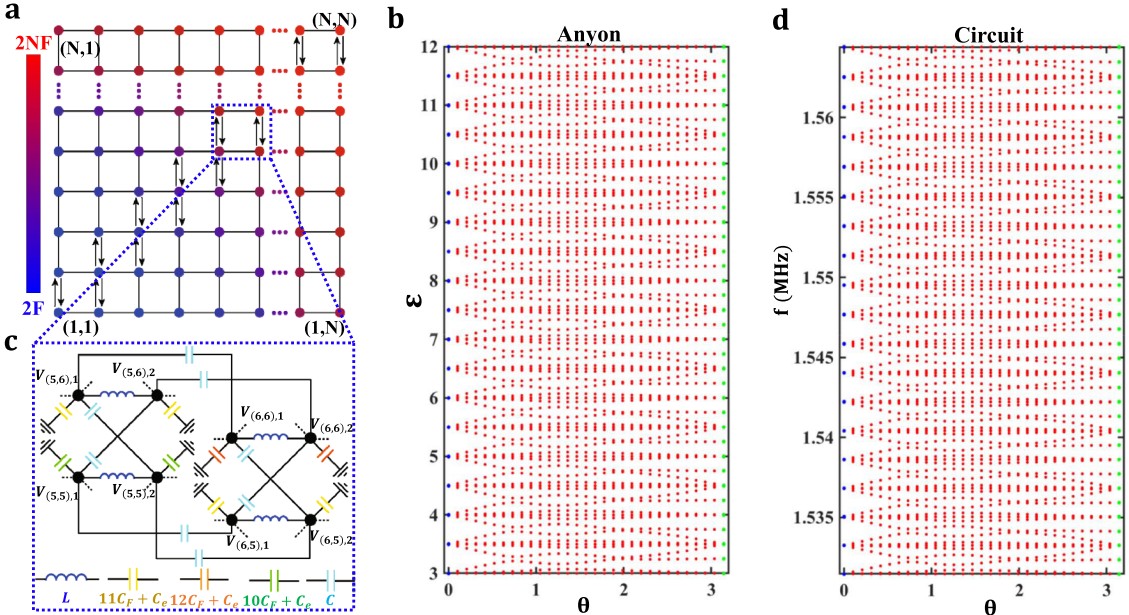

**Fig. 1 Schematic diagram of designed circuit simulators for the Bloch oscillation of a pair of anyons. a** The mapped 2D lattice of the single particle for simulating the 1D two-anyon effect in the absence of on-site interaction under an external forcing. The color represents the value of the on-site potential related to the external forcing. The arrow corresponds to the hopping rate with a complex phase $e^{\pm i\theta}$. **c** Schematic diagram for a part of the designed circuit simulator with $\theta = \pi$ corresponding to four lattice sites enclosed by the blue dashed block in panel a. A pair of circuit nodes belonging to a single site are connected by the blue inductor $L$. The yellow, orange, green capacitors correspond to grounding capacitors at sites of (6, 5) or (5, 6), (6, 6) and (5, 5), respectively. The cyan capacitor marks the connection capacitor between circuit nodes at different sites. **b, d** Calculated eigen-energies of two anyons and eigen-frequencies of the circuit simulator as a function of the statistical angle $\theta$. The blue and green dots correspond to eigen-frequencies of two-boson and two-pseudofermion circuit simulators.

makes the BO frequency of two pseudofermions become half of that for two non-interacting bosons. Although the statistic-induced halving of the BO frequency is very interesting, to date, the experimental observation of such an exotic effect is still lacking even using the potential artificial structures[50–52].

Based on the similarity between the circuit Laplacian and lattice Hamiltonian[53–70], electric circuits can be used as an extremely flexible platform to fulfill the above mapped 2D lattice with different statistical angles. Figure 1c illustrates the schematic diagram for a part of the designed circuit simulator with $\theta = \pi$, which corresponds to four lattice sites enclosed by the blue dashed block in Fig. 1a. Here, a pair of circuit nodes connected by the inductor $L$ are considered to form an effective site in the 2D lattice model. The voltages at these two nodes are marked by $V_{(m,n),1}$ and $V_{(m,n),2}$, which are suitably formulated to form a pair of pseudospins $V_{\uparrow(m,n),\downarrow(m,n)} = (V_{(m,n),1} \pm V_{(m,n),2})/\sqrt{2}$. To simulate the real-valued hopping rate, two capacitors (the capacitance equals to $C$) are used to directly link adjacent nodes without a cross. For the realization of the hopping rate with a phase ($e^{\pm i\pi}$), two pairs of adjacent nodes are cross-connected via $C$. Position-dependent capacitors $(m + n)C_F$ are used for grounding to simulate the spatially modulated on-site potential induced by the external forcing. Moreover, the extra capacitor $C_e$, which is crucial for the achievement of anyonic BOs in the circuit networks (demonstrated below), is also added to connect each circuit node to the ground. Through the appropriate setting of grounding and connecting, the circuit eigen-equation can be derived as:

$$(f_0^2/f^2 - 4 - C_e/C)V_{\downarrow,mn} = -e^{-i\pi(\delta_{m,n}+\delta_{m,n+1})}V_{\downarrow,m(n+1)}$$
$$- e^{i\pi(\delta_{m,n}+\delta_{m+1,n})}V_{\downarrow,m(n-1)} - V_{\downarrow,(m+1)n}$$
$$- V_{\downarrow,(m-1)n} + (m+n)\left(\frac{C_F}{C}\right)V_{\downarrow,mn},$$
(5)

where $f$ is the eigen-frequency ($f_0 = 1/2\pi\sqrt{CL/2}$) of the designed circuit, and $V_{\downarrow,(m,n)} = (V_{(m,n),1} - V_{(m,n),2})/\sqrt{2}$ represents the voltage of pseudospin at the circuit node $(m, n)$. Details for the derivation of circuit eigenequations are provided in Supplementary Note 1. It is shown that the eigen-equation of the designed electric circuit possesses the same form as Eq. (4). In particular, the probability amplitude for the 1D two-pseudofermion model $c_{mn}$ is mapped to the voltage of pseudospin $V_{\downarrow,(m,n)}$ at the circuit node $(m, n)$. The eigen-energy ($\varepsilon$) of two anyons is directly related to the eigen-frequency ($f$) of the circuit as $\varepsilon = f_0^2/f^2 - 4 - C_e/C$, with other parameters being $J = 1$ and $F = C_F/C$. It is worth noting that the method for designing the circuit simulator is applicable to other statistical angles $\theta = \frac{v}{o}\pi$ ($v$ and $o$ are integers), where the complex hopping rate $Je^{\pm i\frac{v}{o}\pi}$ could be realized by suitably braiding the connection pattern of $o$ adjacent circuit nodes in a single lattice site[53,54]. In this case, the relationship between $\varepsilon$ and $f$ with different values of $\theta$ remains the same. We note that the simulation of anyons by designed circuit networks could be intuitively understood as follows. To exchange locations of two anyons, the first anyon should tunnel from the original position (the $m$th site) to the position of the second anyon initially located (the $n$th site), that is from $c_{mn}$ to $c_{nn}$. Then, the second anyon should also move from its original position to the position of the first anyon originally occupied, corresponding to that from $c_{nn}$ to $c_{nm}$. In this case, the effective amplitude for the exchange of two anyons could be expressed by the product of hopping amplitudes in these two processes, and an associated phase factor $e^{\pm i\theta}$ related to the particle statistic should appear. To ensure the appearance of a statistic-related phase factor $e^{\pm i\theta}$, the hopping amplitudes at the diagonal must be $e^{\pm i\theta}$ along one axis.

To analyze the behavior of BOs in the circuit simulator with respect to $\theta$, eigen-frequencies of the designed circuit as a function of the statistical angle $\theta$ should be calculated. The parameters are set as $C = 10$ pF, $C_e = 2$ nF, $L = 10$ µH, and

$C_F = 5$ pF. Due to the nonlinear relationship between the eigenfrequency of the circuit simulator and the eigen-energy of two anyons ($f = f_0/(\varepsilon + 4 + C_e/C)^{1/2}$), the frequency-spectrum of the designed circuit should deviate from the energy-spectrum of two anyons. In fact, such a deviation could be eliminated by setting an extremely large grounding capacitor $C_e$ (see Supplementary Note 2 for details). As shown in Fig. 1d, the frequency-spectrum related to eigen-energies in the range of $(6\,F,\,24\,F)$ is plotted, where the excitation frequency (1.56 MHz) used in simulations and measurements (discussed below) is located within this frequency range. In this case, we can see that the evolution of eigen-frequencies for the circuit simulator versus the statistical phase $\theta$ possesses the same trend as that of the two-anyon eigen-energy. In particular, the nearly equalspaced frequency-spectrum of the designed circuit with $\theta = 0$ ($\triangle f_B \approx 1.863$ kHz) exists, which is analogous to the Wannier-Stark ladder of a pair of non-interacting bosons. Additionally, the frequency-spectrum of the designed circuit for pseudofermions is equal-spaced with the spacing being $\triangle f_f \approx 0.931$ kHz, which is nearly half of $\triangle f_B$. With such a good correspondence between the frequency-spectrum of the designed 2D circuit and the 1D two-anyon model, the behavior of the quantum statistics-dominated BOs should be effectively implemented by the designed electric circuit.

**Numerical results of simulating anyonic Bloch oscillations in electric circuits.** In this part, using the designed circuit simulator with $23 \times 23$ node pairs (corresponding to the 1D two-anyon model with $N = 23$ lattice sites), we numerically simulate the behavior of BOs for two non-interacting anyons with statistical angles being $\theta = 0$ and $\theta = \pi$. Here, the values of $C$, $C_F$, $C_e$ and $L$ are taken as 10 pF, 5 pF, 2 nF, and 10 μH (the same as those used in Fig. 1d). To illustrate the frequency-spectra, as shown in Fig. 2a and b, we calculate the sum of impedance for fifteen diagonal nodes (from (5,5) to (19,19)) in two-boson and two-pseudofermion circuit simulators, respectively. It is clearly shown that various equally spaced impedance peaks appear in the central frequency-domain, manifesting the existence of Wannier-Stark spectra of our designed circuit simulators. The small deviation at low- and high-frequency ranges is due to the finite-size effect, which makes boundary modes be excited in addition to bulk states by circuit nodes far from the center (see Supplementary Note 3 for details). It is noted that the frequency-spacing of two adjacent impedance peaks for the bosonic simulator is two times that for two pseudofermions, which is consistent with the calculated frequency-spectra in Fig. 1d.

To clearly illustrate the statistics-dependent BO in our designed electric circuits, we perform time-domain simulations of voltage dynamics using LTSpice software. First, we focus on the circuit simulator of two non-interacting bosons. Here, the excitation

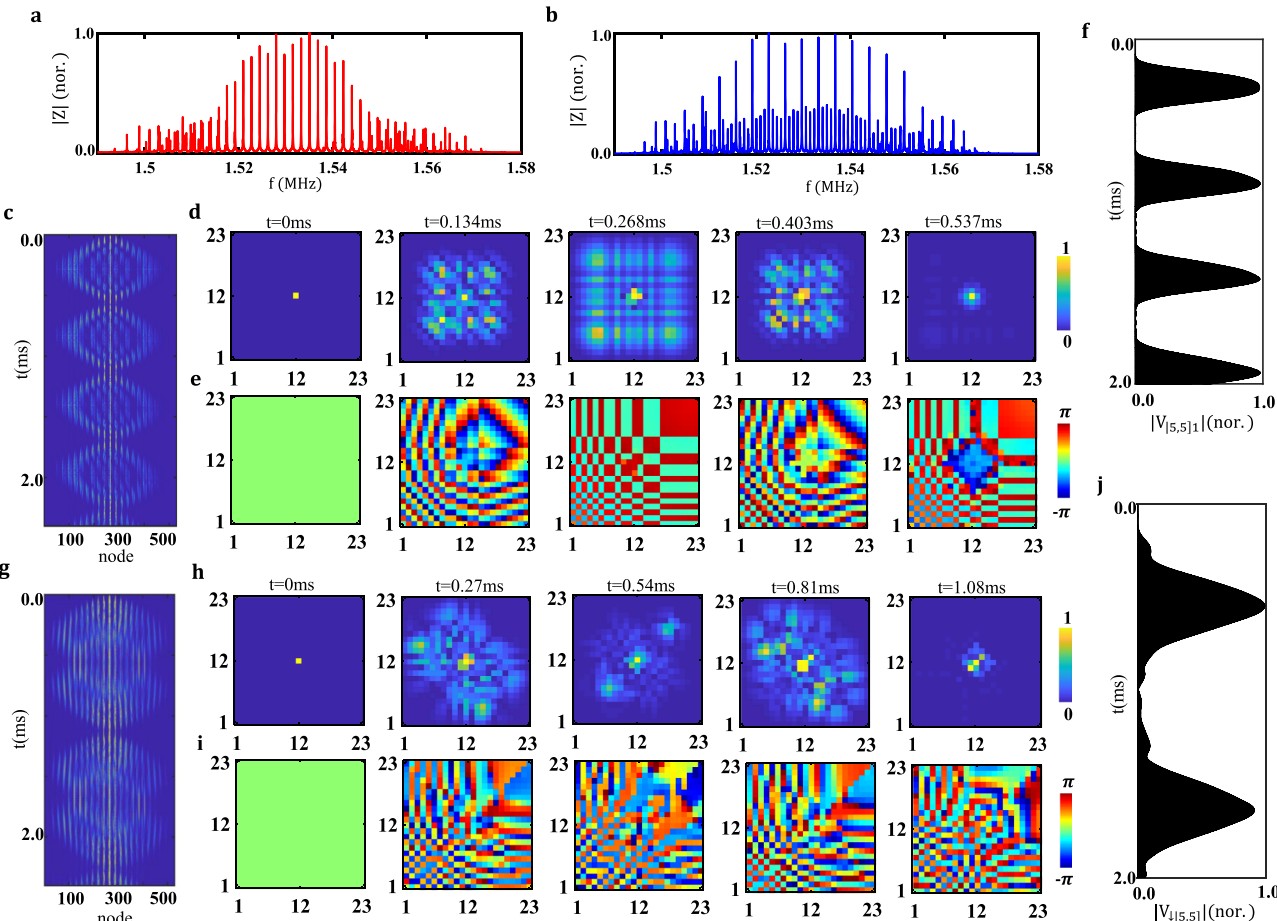

**Fig. 2 Numerical results for simulating anyonic Bloch oscillations in electric circuits. a, b** The sum of impedance for fifteen diagonal nodes (from (5, 5) to (19, 19)) of circuit simulators for two bosons (the red line) and two pseudofermions (the blue line), respectively. The time-dependent evolution of voltage signals at all nodes in the 2D circuit simulator with $\theta = 0$ for (**c**) and $\theta = \pi$ for (**g**). The 2D distributions of voltage amplitude and phase at different times in (**d**) and (**e**) for $\theta = 0$, and in (**h**) and (**i**) for $\theta = \pi$, where subplots from left to right correspond to results with increased times. The simulated voltage signals at the node (5, 5) of 2D circuits with $\theta = 0$ for (**f**), and $\theta = \pi$ for (**j**).

frequency is set as $f = 1.56$ MHz, and the central circuit node is excited by $V_{(12,12),1} = V_0 e^{i2\pi ft}$. Figure 2c displays the time-dependent evolution of $|V_{[m,n],1}(t)|^2$ at all nodes, where the circuit node $(m, n)$ is labeled by $(m - 1)N + n$. We connect a suitable capacitor between the excited circuit node and the voltage source to ensure that the periodic oscillation could also appear at the excited node $(12, 12)$. In this case, although the large occupation exists at the center site (the excited site), it is not a constant but also exhibits periodic oscillation. Amplitude (normalized to the maximum) and phase distributions of voltage signals at selected times with equal intervals (in the first period) are presented in Fig. 2d and e. We can see that voltages at symmetric circuit nodes $(m, n)$ and $(n, m)$ are always the same, being consistent with the commutation relation of two bosons. In addition, it is shown that the voltage displays the periodic breathing dynamics, indicating the appearance of BOs. The time-dependent voltage signals at a selected circuit node ($m = 5$, $n = 5$) is further calculated, as presented in Fig. 2f. We can see that the revival of voltage is clearly verified with a single-site resolution. The oscillation period could be obtained by calculating the time difference between two adjacent voltage maxima of a circuit node. In this case, the associated oscillation period at the circuit node $(5, 5)$ is ~0.548 ms, which is nearly consistent with the period $T_B = 1/\triangle f_B = 0.537$ms predicted by the frequency-spectrum in Fig. 1d. By calculating the period on many other representative circuit nodes, the period error is in the range of $[-0.01$ ms, $0.012$ ms$]$.

Next, we calculate the BO of two non-interacting pseudofermions by the designed electric circuit. The excitation frequency is also set as 1.56 MHz, which is located in the equally spaced region of the frequency-spectrum at $\theta = \pi$. The voltage pseudospin can be suitably excited by setting the input signal as $V_{(12,12),1} = V_0 e^{i2\pi ft}$, $V_{(12,12),2} = -V_0 e^{i2\pi ft}$. The time-dependent evolution of $|V_{\downarrow,[m,n]}(t)|^2$ at all circuit nodes is displayed in Fig. 2g. Figure 2h and i shows the normalized amplitude and phase distributions of the voltage pseudospin at different times in the first period. We can see that the phase difference between a pair of circuit nodes located at $(m, n)$ and $(n, m)$ always equals to $\pi$ (except for the initial time), which is consistent with the requirement of anyonic commutation relation for pseudofermions. Moreover, the calculated voltage signal of $|V_{\downarrow,[5,5]}(t)|$ is presented in Fig. 2j. It is shown that the periodic breathing dynamics of the voltage pseudospin could also appear in the circuit simulator for two pseudofermions. The calculated period of BOs is approximately 1.08 ms (with the error being $[-0.0068$ ms, $0.013$ ms$]$), being consistent with $T_f = 1/\triangle f_f = 1.074$ms predicted by the frequency-spectrum in Fig. 1d. Moreover, compared with the results of the bosonic circuit simulator, we find that the BO frequency in the two-boson circuit simulator is almost twice that in the two-pseudofermion simulator. This is in accord with the theoretical results in the two-anyon lattice model (see Supplementary Note 4 for details).

It is worth noting that the above results only focus on the two-boson/two-pseudofermion models at a fixed excitation frequency of 1.56 MHz and constant values of $C_F = 5$ pF and $C_e = 2$ nF. In Supplementary Note 5, we also simulate anyonic BOs by our designed electric circuits with different excitation frequencies, external forces and grounding capacitors $C_e$. It is shown that the smaller the external force is, the larger the oscillation period and amplitude become. Moreover, we find that the more ideal BO could be realized with a larger value of $C_e$, which could make the frequency-spectra of circuit simulators become more equally spaced than the ideal Wannier-Stark spectrum.

**Experimental observation of anyonic Bloch oscillations in electric circuits**. To experimentally observe the anyonic BOs, the

designed circuit simulators are fabricated, where the corresponding parameters are the same as those used in Fig. 2. A photograph image of the circuit sample is presented in Fig. 3a, and enlarged views of the front and back sides are plotted in the right insets. Here, a single printed circuit board (PCB), which contains $23 \times 23$ node pairs, is applied to the circuit. It is noted that our fabricated circuit simulator could perform the BO of two anyons with $\theta = 0$ ($\theta = \pi$) when the switches (enclosed by white blocks) located around the diagonal line of the sample are opened (closed). This is because these switches could change the connection pattern between adjacent circuit nodes from direct connections to cross-connections. In this case, if two pairs of adjacent circuit nodes are directly (cross) connected through the capacitor $C$ (framed by red circles), the hopping rate without (with) a phase factor $e^{\pm i\pi}$ could be realized, which is required for the two-boson (two-pseudofermion) circuit simulator. Moreover, the position-dependent grounding capacitors $(m + n)C_F$ (framed by blue circles) are used to implement the external forcing. The inductor $L$ and grounding capacitor $C_e$ are enclosed by the pink and green frames, respectively. Additionally, the tolerance of the circuit elements is less than 1% to avoid the detuning of circuit responses. Details of the sample fabrication are provided in Methods.

Firstly, as shown in Fig. 3b and c, the summed impedance of fifteen diagonal nodes (from $(5, 5)$ to $(19, 19)$) are measured in the fabricated two-boson and two-pseudofermion circuit simulators using a Wayne Kerr precision impedance analyzer. We can see that the equal-spaced impedance peaks also exist in experiments. Compared with simulation results in Fig. 2a and b, the larger width of measured impendence peaks results from the lossy effect in the fabricated circuit. The frequency-spacing of two adjacent impedance peaks for the bosonic circuit simulator is still nearly two times that for the pseudofermion circuit simulator (three little peaks exist between two large peaks shown in the inset), indicating that our fabricated circuits could indeed exhibit Wannier-Stark spectra of two bosons and two pseudofermions.

Then, we measure the temporal dynamics of the fabricated electric circuit with $\theta = 0$ (open switches), where a circuit node is excited by $V_{(12,12),1} = V_0 e^{i2\pi ft}$ ($V_0 = 1$V) with $f = 1.56$ MHz. The measured amplitude and phase distributions of the voltage signal at different times (in the first period) are plotted in Fig. 3d. We can see that the symmetric voltage distribution exists. The small derivation should result from the weak disorder effects in the circuit (see Supplementary Note 6). Moreover, Fig. 3e displays the measured voltage signal at $(5, 5)$ in the time-domain. It is clearly shown that the damped BO appears. Based on the same method of obtaining the BO period in simulations, the measured damped BO period is ~0.535 ms (with the error being $[-0.015$ ms, $0.017$ ms$]$), which is consistent with the simulation. The decay of the revival voltage is due to the large lossy effect, resulting from the resistive loss of linking wires and the finite Q-factor of the applied inductor. To fit the strength of loss in the fabricated circuit, we calculate the voltage dynamics with different series resistances of inductors (see Supplementary Note 7). In this case, we can deduce that the effective series resistance of inductance in the fabricated circuit is approximately 50 $m\Omega$.

Finally, we turn to the fabricated circuit with $\theta = \pi$ (close switches), and the excitation signal is set as $V_{(12,12),1} = V_0 e^{i2\pi ft}$, $V_{(12,12),2} = -V_0 e^{i2\pi ft}$. As shown in Fig. 3f, the measured amplitude and phase distributions of $V_{\downarrow,[m,n]}(t)$ at different times in the first period are presented. It is shown that the nearly asymmetric phase distribution is observed. Similar to the case of two bosons, the small derivation should result from the disorder effects in the circuit sample. In addition, Fig. 3g displays the measured voltage signal of $|V_{\downarrow,[5,5]}(t)|$. We can see that the

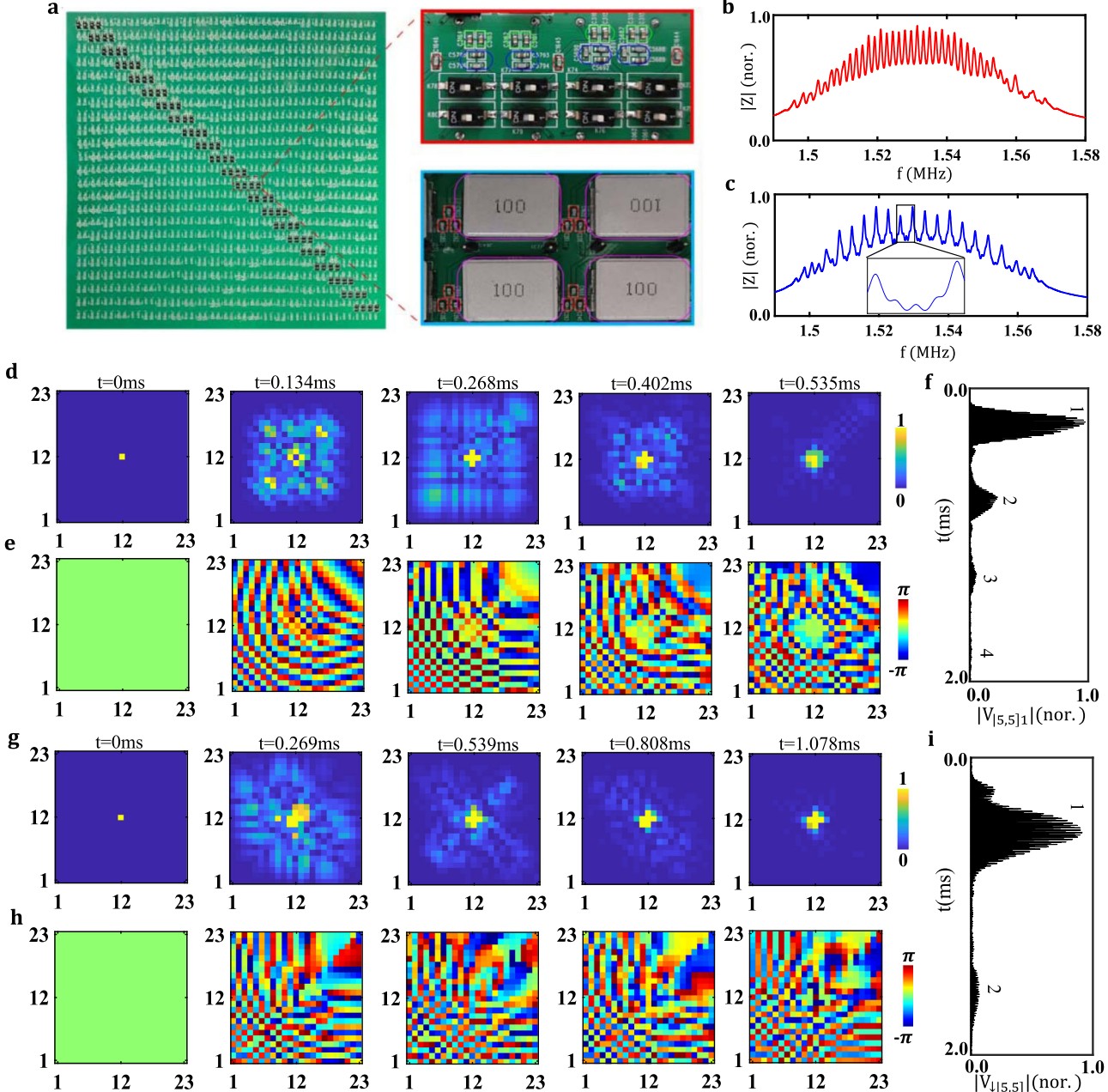

**Fig. 3 Experimental results for observing anyonic Bloch oscillations. a** Photograph images of the fabricated circuit simulator. The enlarged views for the front and back sides are shown in right insets. The switches are enclosed by white blocks around the diagonal line. Two pairs of adjacent circuit nodes are connected through the capacitor $C$ framed by red circles). The position-dependent grounding capacitors $(m + n)C_F$ are framed by blue circles. The inductor $L$ and grounding capacitor $C_e$ are enclosed by pink and green frames, respectively. **b, c** The measured sum of impedance for fifteen diagonal nodes (from (5, 5) to (19, 19)) of circuit simulators for two bosons (the red line) and two pseudofermions (the blue line), respectively. The measured distributions of voltage amplitude and phase at different times in (**d**) and (**e**) for $\theta = 0$, and in **g** and **h** for $\theta = \pi$. The measured voltage signals of the circuit node (5, 5) with $\theta = 0$ for (**f**), and $\theta = \pi$ for (**i**).

damped oscillation period is ~1.078 ms (with an error being [−0.021 ms, 0.016 ms]), which is also consistent with the simulated result. Compared to the measured period with $\theta = 0$, we note that the BO frequency related to a pair of pseudofermions is also half of that for two bosons. Moreover, similar to the bosonic circuit simulator, the significant decay of the voltage signal results from the large lossy effect, where the fitted series resistance of inductances is about 50 $m\Omega$.

It is worthy to note that the loss in fabricated circuits could be mapped to the dissipation rate of the 1D anyonic lattice model. To clarify the influence of losses on the anyonic BOs, we extend

the two-anyon lattice model in Eq. (1) to contain the intrinsic dissipation rate[71] (see Supplementary Note 8 for details). In this case, similar to the measured voltage dynamics, the damped BOs of two anyons also appear.

## Discussion

We note that the above-designed $LC$ circuit possesses the identical stationary eigen-equation with that for the 1D two-anyon system. With the advantage of diversity and flexibility for circuit elements, we can design another kind of electric circuit, which is based on resistances and capacitances, to precisely match the

time-dependent Schrödinger equation of two pseudofermions and two bosons. In this case, the BOs dominated by quantum statistics can also be observed in the designed *RC* circuit. Detailed results are given in Supplementary Note 9.

In addition, it is worthy to stress that the near-perfect Wannier-Stark spectrum could also appear at other statistical angles (besides $\theta = 0$ and $\theta = \pi$) under a suitable value for the ratio of the external forcing to the hopping rate (*F/J*), where the corresponding period of the BO could become three times of that for two bosons (see Supplementary Note 10 for details). Such a BO dominated by particle statistics beyond bosons and pseudofermions could also be simulated by designed RLC circuit networks combined with a negative impedance converter with current inversion[72].

In conclusion, we have experimentally demonstrated that electric circuits can be used as a flexible simulator to investigate the statistics-dominated BO, where the oscillation frequency of two pseudofermions is half of that for two bosons in the absence of on-site interaction. Using the exact mapping of two anyons in the external forcing to modes of designed circuit lattices, the periodic breathing dynamics of voltage in circuit simulators with $\theta = 0$ and $\theta = \pi$ have been observed. Our proposal could provide a flexible platform to further investigate and visualize many interesting phenomena related to particle statistics and other exotic few-particle physics. With the flexibility that the connection and grounding of circuit nodes are allowed in any desired way free from constraints of locality and dimensionality, the anyonic physics existing in the lattice model with nonlocal hopping and interactions (beyond nearest neighbors) could also be achieved. Moreover, by mapping the 1D multiple-anyon model to the higher-dimensional lattice model, the circuit network could also be used to simulate anyonic physics with more particles. Furthermore, including nonreciprocal and non-Hermitian elements in the circuit network, the novel behavior induced by the interplay between the non-Hermitian effect[73] and the quantum correlation can be investigated. In addition, electric circuits are easily fabricated using existing chip manufacturing technology, making the achievable number of circuit nodes become extremely increased. Such an electric chip could make the designed circuit simulator implement much more complex anyonic physics, such as statistically induced phase transitions and statistics-related topological phases. Finally, the designed circuit simulator could also give a way to manipulate the electronic signals with exotic behaviors.

## Methods

**Sample fabrications and circuit signal measurements**. We exploit the 2D electric circuits by using PAD program software, where the PCB composition, stack-up layout, internal layer, and grounding design are suitably engineered. Here, the well-designed 2D PCB possesses six layers, containing the top layer, the bottom layer, two mid-layers, and two internal planes, to suitably arranging circuit elements, linking wires, and the ground setting. It is worth noting that the ground layer should be placed in the gap between any two layers to avoid mutual inductance. Moreover, all PCB traces have a relatively large width (0.75 mm) to reduce the parasitic inductance, and the spacing between electronic devices is also large enough (0.3–0.5 mm) to avert spurious inductive coupling. The SMP connectors are welded on PCB nodes for signal injection and detection. To ensure the realization of BOs in electric circuits, both the tolerance of circuit elements and the series resistance of inductors should be as low as possible. For this purpose, we use a WK6500B impedance analyzer to select circuit elements with high accuracy (the averaged disorder strength is less than 1%) and low losses.

For the measurement of BOs, we use the signal generator (NI PXI-5404) with eight output ports to act as the current source for exciting one/two circuit nodes related to a single lattice site with a constant amplitude and node-dependent initial phases. To ensure that the periodic oscillation could also appear at the excitation node, we connect a suitable capacitor (~100 pF) between the excitation circuit node and the input source. One output of the signal generator (the initial phase is set to 0) is directly connected to one end of the oscilloscope (Agilent Technologies Infiniivision DSO7104B) to ensure an accurate start time. The measured voltage signals are in the range from 0 ms to 2 ms in the time-domain, where 0 ms is defined as the time for the simultaneous signal injection and measurement. We repeat the time-domain measurements of the circuit simulator, which exhibit the same response under repeated excitations, to obtain voltage signals of all circuit nodes. Finally, the

measured real-valued voltage signals are transformed into complex ones based on the Hilbert-transform, which gives the results of phase distributions at different times.

## Data availability
All data are displayed in the main text and Supplementary Information. The data that support the findings of this study are available from the corresponding author upon reasonable request.

## Code availability
The code that supports the plots within this paper is available from the corresponding author upon reasonable request.

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

## Acknowledgements

W.Z., H.Y., H.W., F.D., N.S., X. Zheng, H.S. and X. Zhang are supported by the National Key R&D Program of China under Grant No. 2017YFA0303800 and the National Natural Science Foundation of China (Nos. 91850205 and 61421001. and No.12104041).

## Author contributions

W.Z. finished the theoretical scheme and designed the circuit simulator. H.Y., H.W., F.D., N.S., and X. Zheng finished the experiments with the help of H.S. W.Z. and X. Zhang wrote the manuscript. X. Zhang initiated and designed this research project.

## Competing interests

The authors declare no competing interests.
