## [Peer Review File · Nature Communications]

REVIEWER COMMENTS

Reviewer #1 (Remarks to the Author):

In this manuscript "Observation of anyonic Bloch oscillations ", Zhang et al built an electrical circuit that serves to simulate the Bloch oscillations dynamics of a mathematically equivalent system of interacting anyonic particles. The physical simulation of interacting condensed matter systems, particularly of exotic particles, is one of the major challenges in today's study of condensed matter, and this work serves contributes an advance to this important theme. It is also one of the latest - and most sophisticated - example of how electrical circuits (commonly known as topoelectrical circuits) can be used as reliable and versatile simulators of condensed matter systems, particularly those that are very difficult to simulate with other platforms like metamaterials and cold atoms.

As such, I think this work will gain a lot of attention, and should eventually be published in Nature Communications, especially since it represents a far more sophisticated technical advance, in comparison, to other previous works in Nat Comms i.e. Ref 56.

I will recommend publication after the following improvements have been made:

1. Both the simulation and experiment results show clear periodicity in the voltage signals and thus clear oscillations. However, what is not so clear is how to see that these are really "anyonic"? The authors should compare the oscillations of anyonic and boson systems side-by-side, and particularly compare their frequency spectra.
2. Related to the above, how can one see where the "anyons" live in the circuit? I understand the anyonic statistics is built into the model, but it will be good to justify the physical interpretation of the "anyon", such as to be convinced that the anomalous oscillation period is really due to anyonic statistics.
3. Is the setup a 2D or 3D circuit? It looks like a 2D circuit from fig 3 and fig 1c, but it was mentioned in the methods that there are multiple layers. Any photos showing how the multiple layers are connected?

4. Regarding the presentation, I find the circuit node plots like 2a and 2d confusing. We have a 2D (or 3D?) system, but Figs 2a and 2d only show the nodes numbered arbitrarily from 1 to 500 (or 529?). Hence it is hard to physically understand the qualitative shapes seen in them. Perhaps the authors should instead show some fixed time slices of the measured signals, at least in the main text.

5. Also about the presentation, figures like 1b and 1d are hard to read because many of the dots are very close together. May be better to show a taller figure with dots that are more clearly separated, and with much finer resolution in theta, such that one can follow how the spectrum evolves as theta is varied. Or is it the point that the spectrum is never continuous?

Also, do figs 1b and 1d show exactly the same information, just with different y-axis labels?

6. How are the curves in Figs 3b,c related to the oscillations in figs 2b,c,e,f? They certainly look quite different.

7. Some care may be needed to distinguish "numerical simulations" and "circuit simulations". Perhaps the experimental results/setup can be referred to as "experiments" so as to not confuse them with the (rigorously done) numerical simulations of the circuits?

8. In the supplement section on two bosons and two pseudofermions, INICs are used. Such INIC networks have been implemented in

PRL 126 (21), 215302, even if the eigenvalues are still real.

Is the circuit still Hermitian? And are the different oscillation frequencies then remotely related to nonhermitian mechanisms, for instance in doi.org/10.1038/s42005-020-00417-y ?

9. Last but not least, do check the manuscript for grammatical and spelling errors i.e. "still a great challenge" instead of "still a great challenging" in line 58, also the sentence seems incomplete, starting from "While". In line 59, are newly accessible platforms really "expected" ? In line 271, "except" makes the sentence sound awkward.

Reviewer #2 (Remarks to the Author):

This work studies a model for a pair of particles in which the statistical phase can be varied. The authors demonstrate that the two-particle 1D model can be mapped onto a single-particle 2D lattice. They then demonstrate that this model can be simulated with a suitably-designed circuit. Finally, the authors fabricate the circuit for the cases of bosons and pseudo-fermions and measure voltage oscillations. They observe oscillations in the envelope function matching their simulations and theoretical predictions, providing validation for their approach to simulation of the quantum model.

This is an interesting work which illustrates the power of electronic circuits to model exotic quantum phenomena which are difficult to realize experimentally. The novelty of this work is enhanced by the experimental implementation of the circuit. Thus, I believe this work should be published. However, I have a few questions and suggestions for the authors:

1. The authors should emphasize that the theoretical model they describe starting on line 72 was developed by Longhi and Valle, PRB 85, 165144 (ref. 42).
2. There are additional theoretical works which provide the basis for the 1D anyon model used in this work. I believe it is appropriate for the authors to cite A. Kundu, PRL 83, 1275–1278 and T. Keilman et al, Nat. Comm. 2, 361 which develop the framework for 1D anyons and pseudofermions.
3. Can the authors describe in more detail how the excitation is applied to the simulated and real circuits? Is a short pulse used, or is it a continuous excitation starting from $t = 0$? Additionally, can the authors discuss what this circuit excitation corresponds to physically in the anyon-Hubbard model?
4. It is found that in the physical circuit, the Bloch oscillations decay with time due to losses. Do these losses correspond to any physical processes in the anyon-Hubbard model, i.e. leakage of particles out of the system?

5. In the Supplementary Material S1, there is a parameter C_U , which I did not find mentioned elsewhere in the manuscript. What

is this capacitance?

6. I am not sure that it makes sense to describe the simulated particles as "anyons." Anyons are generally understood to be distinct from bosons and fermions in having a statistical phase different from 0 or π . While the model that the authors employ can be generalized to describe anyons, their simulations and realized circuits have focused on the statistical phases of 0 and π , corresponding to bosons and pseudo-fermions rather than anyons. Thus, a more accurate title might be "Observation of Bloch oscillations for bosons and pseudo-fermions."

Reviewer #3 (Remarks to the Author):

In the manuscript "Observation of Anyonic Bloch oscillations" the authors Zhang et al. discuss a realization of two-particle anyon physics by coupled electric circuits and they observe the effect of statistics through the analysis of Bloch oscillations.

In my opinion, the manuscript has an interesting aim but the actual content is quite modest. In particular, the experimental results are limited to a single plot in Fig. 4 and a single quantity, the amplitude V on a certain element of the circuitry, whereas most of the theoretical ideas are just a repetition of Ref. 42 (Longhi, Della Valle PRB 2012), which are then applied to the circuit platform. Besides, the anyonic Bloch oscillations for $\theta=\pi$ are then relegated to the last figure panel of the manuscript, Fig.4c, and represented by a very small and barely observable peak, which should have been the main highlight of the paper but rather seems a more modest outcome. As an example of more detailed analysis, e.g. for Fig. 4, it would have been useful to quantitatively estimate the losses and show that the measured damping can be explained by a more refined model, if that is possible. Besides, is there no other measurement to show in order to validate or present the results except for the dynamics of a single site? Is there no other measurement beyond the ones already discussed by Longhi et al.? Moreover, I may disagree with the authors' statement (which contains no further motivation) that the anyonic Bloch oscillations are an intractable challenge for quantum/classical platforms as a motivation for their work. I will comment further on this here below.

To summarize: I think that the paper does not contain 1) a significant amount of experimental results 2) a significant amount of innovative content that justifies publication in Nature

Communications. I may change my mind with a substantial improvement of the experimental results and of the theoretical modeling, namely with major improvements of the manuscript and a more in-depth analysis of the system and the physics that they can access.

Here below some further points that the author may consider to address:

1) The authors state that the realization of the effect is an intractable challenge for quantum/classical platforms without providing further motivation. Unless they provide a striking argument, I will strongly disagree with it. As shown in Fig. 1a the lattice can be realized by tailoring certain hopping coefficients near the main diagonal, and in particular by realizing a negative hopping to obtain π -flux on certain plaquettes for the $\theta = \pi$ case. This can easily be realized in photonics (see for example ring resonators Mittal et al. Nature Photonics volume 13, pages 692–696 (2019) or optical waveguides Mukherjee et al. Phys. Rev. Lett. 121, 075502 (2018), Kremer et al. Nature Communications volume 11, Article number: 907 (2020)).

2) The authors write an anyon Hubbard model in Eq. 1 but the onsite U Hubbard interaction is never used in the manuscript. I do not understand why they are therefore talking about an interacting model, when interactions do not play a role in the analysis presented here (but they are used in Ref. 42)

3) The commutation relations in Eq. 2 show that the particles behave as fermions when they are at different sites. This translates, for $\theta = \pi$, to $c_{nm} = -c_{mn}$. Is this condition fulfilled in their results? In principle this is an extra constraint which is not necessarily realized away from the diagonal, unless the modes are projected to the antisymmetric sector. Without this condition in place, the claim that the experiment is observing anyonic statistics for two particles is not true. If one looks at the spectrum, it should contain also states where the antisymmetric constraint is not fulfilled in a certain range. So the question can be rephrased by clarifying if the "wavefunction" is in the right sector when they observe the time dynamics, namely if the corresponding parity is conserved over time. The authors should also estimate how imperfections of the setup may lead to deviations from this. This could be a nice piece of information that makes the analysis less straightforward, for example.

3b) The commutation relations of interest to the authors have been discussed and brought to the audience of synthetic systems by Keilmann et al. Nature Communications volume 2, Article number: 361 (2011), which cites other papers on this topic as well. I suggest the authors to review the reference list accordingly besides citing just Ref. 42.

4) In Fig. 1b-d, what is the reason for plotting the spectrum in that specific range? This is not clear to me and should be motivated in the text.

5) I have not found an explanation of the fact that $\theta = \pi$ anyons have a doubling of Bloch oscillations period. From what I have seen in the manuscript, the authors limit themselves to state that this is the result in Ref. 42. I think this can be clarified and further developed. In general the Wannier Stark spectrum and gaps can be deduced for a single band and the authors could elaborate further in this direction.

6) The role of dissipation is not quantified but only mentioned. If this had a more quantitative analysis, it could explain the experimental results through a more refined theoretical modeling. An example of spectral analysis in the dissipative case is for instance provided by the work "Gorlach et al. Phys. Rev. A 98, 063625 (2018)" for the simulation of two-particle physics. If the single-site damping is understood, the modeling of dissipation could be used to quantify the observation in time displayed in Fig. 3. This analysis would show if dissipation is or is not the only mechanism in place for the observed outcome, perhaps disorder plays also a role. All these details should be explored and clarified a bit more in depth for an experimental paper.

7) In line 207, the authors estimate the oscillation period (0.545ms). More details of this estimate (is it a fit, an extrapolation, a fourier transform?) and of the corresponding error should be presented to the reader. Same comment applies to the experimental results.

8) I have also found a very unsatisfactory concluding paragraph, which does not provide much information about the impact of this work beyond the simple realization of two particle physics described by the Hamiltonian (1).

Response Letter to Reviewers

We are grateful for the constructive comments on this manuscript (NCMMS-21-40151) from all three reviewers.

In the text below, reviewer comments are quoted in blue and followed by our detailed response. We have also revised the manuscript and the Supplemental Materials based on the reviewer comments, and these updates are highlighted in red in those files. In the text below, these updates are also highlighted in *Italics*.

Response to comments of the reviewer #1

In this manuscript "Observation of anyonic Bloch oscillations", Zhang et al built an electrical circuit that serves to simulate the Bloch oscillations dynamics of a mathematically equivalent system of interacting anyonic particles. The physical simulation of interacting condensed matter systems, particularly of exotic particles, is one of the major challenges in today's study of condensed matter, and this work serves contributes an advance to this important theme. It is also one of the latest- and most sophisticated- example of how electrical circuits (commonly known as topoelectrical circuits) can be used as reliable and versatile simulators of condensed matter systems, particularly those that are very difficult to simulate with other platforms like metamaterials and cold atoms.

As such, I think this work will gain a lot of attention, and should eventually be published in Nature Communications, especially since it represents a far more sophisticated technical advance, in comparison, to other previous works in *Nat. Comms.* i.e. Ref. 56.

I will recommend publication after the following improvements have been made.

Reply: We would like to thank the reviewer for the positive evaluation and valuable suggestions of our work. In the following, we will give a detailed response to all points proposed by the reviewer.

1). Both the simulation and experiment results show clear periodicity in the voltage signals and thus clear oscillations. However, what is not so clear is how to see that these are really "anyonic"? The authors should compare the oscillations of anyonic and boson systems side-by-side, and particularly compare their frequency spectra.

Reply: We would like to thank the reviewer for the comment. Following the review's suggestion, to compare the associated frequency-spectra, we calculate and measure the sum of impedance for fifteen diagonal nodes (from (5,5) to (19,19)) in two-boson and two-pseudofermion circuit simulators, respectively, as presented in Figs. R1a and R1b. It is clearly shown that equal-spaced impedance peaks appear in the central frequency range, manifesting the existence of Wannier-Stark spectra of our designed circuit simulators. The larger width of measured impedance peaks results from the lossy effect in the fabricated sample. A little deviation at the low- and high-frequency ranges is due to the finite size effect, which makes boundary modes could also be excited in addition to bulk states at circuit nodes far from the center (the (12, 12) node). In S3 of the Supplementary Materials, we numerically consider larger circuit simulators, and find that more ideal equal-spaced impedance peaks could appear in a wider

frequency range. In addition, being consistent with the frequency-spectra in Fig. 1d, the frequency spacing of two adjacent impedance peaks for the bosonic simulator is two times of that for two pseudofermions, which makes the BO frequency of two pseudofermions become half of that for two noninteracting bosons.

Fig. R1. (a) and (b) The calculated and measured sum of impedance for fifteen diagonal nodes (from (5,5) to (19,19)) of circuit simulators for two bosons and two pseudofermions, respectively.

Fig. R2. (a) and (b) Numerical results for the distribution of voltage amplitude and phase at different times (in a period) for two-boson and two-pseudofermion circuit simulators.

In addition, to further illustrate the anyonic properties, distributions of voltage amplitude and phase at different times with equal intervals (in a period) are also proposed in Fig. R2 for numerical results and Fig. R3 for experimental results, where sub-plots from left to right correspond to results with increased times. Here, the voltage amplitudes are normalized by the maximum voltage signal. We can see that calculated voltages at circuit nodes (m, n) and (n, m) are always the same in the two-boson simulator, which is matched to the commutation relation of two bosons $c_{mn} = c_{nm}$. Additionally, the calculated phase difference between circuit nodes (m, n) and (n, m) in the two-pseudofermion simulator always equals to π (except for the initial time), that is also consistent to the requirement of the commutation relation for two pseudofermions $c_{mn} = e^{i\pi}c_{nm}$. The little derivation of measured averaged phase differences results from the weak disorder effects in the fabricated circuit simulator. Above results combined with voltage dynamics are presented side-by-side in the main text to further illustrate the anyonic properties.

Fig. R3. (a) and (b) Experimental results for the distribution of voltage amplitude and phase at different times (in a period) for two-boson and two-pseudofermion circuit simulators.

Action taken:

- In the revised manuscript, we have added simulated and measured distributions of voltage amplitude and phase at different times in Fig. 2 and Fig. 3.
- In the revised manuscript, we have added simulated and measured impedance responses of two-boson and two-pseudofermion circuit simulators in Fig. 2 and Fig. 3.

- In the S3 of revised Supplementary Materials, we have added numerical results on the influence of finite size effect on impedance spectra of circuit simulators.
- In the revised manuscript, we have added the following discussions in pages 7 and 10 on the impedance spectra of designed electric circuits 1): *“To illustrate the frequency-spectra, as shown in Figs. 2a and 2b, we calculate the sum of impedance for fifteen diagonal nodes (from (5,5) to (19,19)) in two-boson and two-pseudofermion circuit simulators, respectively. It is clearly shown that various equally spaced impedance peaks appear in the central frequency domain, manifesting the existence of Wannier-Stark spectra of our designed circuit simulators. The little deviation at low- and high-frequency ranges is due to the finite size effect, which makes boundary modes be excited in addition to bulk states by circuit nodes far from the center (see S3 in the Supplementary Materials for details). It is noted that the frequency spacing of two adjacent impedance peaks for the bosonic simulator is two times of that for two pseudo-fermions, that is consistent with the calculated frequency-spectra in Fig. 1d.”* 2): *“Firstly, as shown in Figs. 3b and 3c, the summed impedance of fifteen diagonal nodes (from (5,5) to (19,19)) are measured in the fabricated two-boson and two-pseudofermion circuit simulators using a Wayne Kerr precision impedance analyzer. We can see that the equal-spaced impedance peaks also exist in experiments. Compared with simulation results in Figs. 2a and 2b, the larger width of measured impedance peaks results from the lossy effect in the fabricated circuit. The frequency-spacing of two adjacent impedance peaks for the bosonic circuit simulator is still nearly two times of that for the pseudofermion circuit simulator (three little peaks exist between two large peaks shown in the inset), indicating that our fabricated circuits could indeed exhibit the Wannier-Stark spectra of two bosons and two pseudofermions.”*
- In the revised manuscript, we have added following discussions in page 7 to illustrate the simulated 2D voltage distributions for two bosons: *“Fig. 2c displays the time-dependent evolution of $|V_{[m,n],1}(t)|^2$ at all nodes, where the circuit node (m, n) is labeled by $(m-1)N+n$. Amplitude (normalized to the maximum) and phase distributions of voltage signals at selected times with equal intervals (in the first period) are presented in Figs. 2d and 2e. We can see that voltages at symmetric circuit nodes (m, n) and (n, m) are always the same, being consistent with the commutation relation of two bosons.”*
- In the revised manuscript, we have added following discussions in page 8 to illustrate the simulated 2D voltage distributions for two pseudofermions: *“Figs. 2h and 2i show normalized amplitude and phase distributions of the voltage pseudospin at different times in the first period. We can see that the phase difference between a pair of circuit nodes located at (m, n) and (n, m) always equals to π (except for the initial time), that is consistent with the requirement of anyonic commutation relation for pseudofermions.”*
- In the revised manuscript, we have added following discussions in page 11 to illustrate the measured 2D voltage distributions for two bosons: *“The measured amplitude and phase distributions of voltage signal at different times (in the first period) are plotted in Fig. 3d. We can see that the symmetric voltage distribution exists. The little derivation should result from the weak disorder effects in the circuit (see S6 of the Supplementary Materials).”*
- In the revised manuscript, we have added the following discussion in page 11 to illustrate the measured 2D voltage distributions for two pseudofermions: *“As shown in Fig. 3f, the measured amplitude and phase distributions of $V_{[m,n]}(t)$ at different times in the first period are presented. It is shown that the nearly asymmetric phase distribution is observed. Similar to the case of two bosons, the little derivation should result from the disorder effects in the circuit sample.”*

2). Related to the above, how can one see where the "anyons" live in the circuit? I understand the anyonic statistics is built into the model, but it will be good to justify the physical interpretation of the "anyon", such as to be convinced that the anomalous oscillation period is really due to anyonic statistics.

Reply: We would like to thank the reviewer for the comment. As shown in Figs. R1-R3 of the above comment, the measured/calculated impedance spectra and voltage dynamics in designed circuit simulators could clearly present the Wannier-Stark spectra and symmetries of voltage distribution determined by the anyonic statistic, respectively. These results give a clear manifestation of the anyonic properties in circuits.

On the other hand, the eigen-equation of two anyons (in Eq. 4) hopping on the 1D lattice under the external force could be precisely obtained by solving the Hamiltonian in the two-anyon Fock space with the limitation of anyonic commutation relations. The obtained eigen-equation describes the coupling behavior between two-anyon states c_{nm} (the first anyon is located at the n th site, and the second anyon is located at the m th site). It is noted that the eigen-equation of two anyons in the 1D lattice can be mapped to the eigenequation of designed 2D circuit. In this case, the influence of statistical angle is reflected by the effective coupling strength on the diagonal of designed circuit, where a statistic-related phase factor $e^{\pm i\theta}$ exists along one axis. Such a statistic-related coupling amplitude on diagonals of the mapped 2D circuits could be intuitively understood as follows. To exchange locations of two anyons, the first anyon should tunnel from the original position (the m th site) to the position of the second anyon initially located (the n th site), that is from c_{mn} to c_{nm} . Then, the second anyon should also tunnel from its original position (the n th site) to the position of the first anyon originally occupied (the m th site), corresponding to that from c_{nn} to c_{mm} . In this case, the effective amplitude for the exchange of two anyons could be expressed by the product of hopping amplitudes in these two processes. When these hoppings are finished, two anyons are exchanged, and an associated phase factor $e^{\pm i\theta}$ related to the particle statistic should appear. To ensure the appearance of a statistic-related phase factor $e^{\pm i\theta}$, the hopping amplitudes at the diagonal must be $e^{\pm i\theta}$ along one axis, as shown in Fig. 1a.

Action taken:

- In the revised manuscript, we have added the following discussion in page 6 to provide an intuitive understanding of the effective lattice model of two anyons: *“We note that the simulation of anyons by designed circuit networks could be intuitively understood as follows. To exchange locations of two anyons, the first anyon should tunnel from the original position (the m th site) to the position of the second anyon initially located (the n th site), that is from c_{mn} to c_{nm} . Then, the second anyon should also move from its original position to the position of the first anyon originally occupied, corresponding to that from c_{nn} to c_{mm} . In this case, the effective amplitude for the exchange of two anyons could be expressed by the product of hopping amplitudes in these two processes, and an associated phase factor $e^{\pm i\theta}$ related to the particle statistic should appear. To ensure the appearance of a statistic-related phase factor $e^{\pm i\theta}$, the hopping amplitudes at the diagonal must be $e^{\pm i\theta}$ along one axis.”*

3). Is the setup a 2D or 3D circuit? It looks like a 2D circuit from fig. 3 and fig. 1c, but it was mentioned in the methods that there are multiple layers. Any photos showing how the multiple layers are connected?

Reply: We are sorry for the confusing statement. Our designed circuit simulator is actually 2D, where the voltage at the circuit node (m, n) in 2D space is directly mapped to the probability amplitude of two-anyon state with one located at the m th site and the other located at the n th site in the 1D lattice. Multiple

layers mentioned in methods correspond to the design detail for realizing the 2D PCB of circuit. Specifically, the well-designed 2D PCB possesses six layers, containing the top layer, the bottom layer, two mid-layers, and two internal planes, to suitably arrange circuit elements, linking wires and the ground setting.

Action taken:

- In the revised manuscript, we have added the following discussion to methods: “*Here, the well-designed 2D PCB possesses six layers, containing the top layer, the bottom layer, two mid-layers, and two internal planes, to suitably arrange circuit elements, linking wires and the ground setting.*”.

4). Regarding the presentation, I find the circuit node plots like 2a and 2d confusing. We have a 2D (or 3D?) system, but Figs. 2a and 2d only show the nodes numbered arbitrarily from 1 to 500 (or 529?). Hence it is hard to physically understand the qualitative shapes seen in them. Perhaps the authors should instead show some fixed time slices of the measured signals, at least in the main text.

Reply: We would like to thank the reviewer for the suggestion. Firstly, we want to clarify that each 2D circuit node (m, n) is labeled by $(m-1)N+n$ in Figs. 2a and 2b (Figs. 2c and 2g in the revised manuscript). To physically understand the qualitative shapes of voltage signals, following reviewer’s suggestion, the simulated and measured 2D voltage distributions of two-boson and two-pseudfermion circuit simulators at different times (within a period) are added in Fig. 2 and Fig. 3 of the main text (also in Fig. R2 and Fig. R3). In this case, the detailed voltage distributions within a BO period are clearly illustrated.

Action taken:

- In the revised manuscript, we have added simulated and measured results for the 2D voltage distributions at different times in Fig. 2 and Fig. 3.

5). Also, about the presentation, figures like 1b and 1d are hard to read because many of the dots are very close together. May be better to show a taller figure with dots that are more clearly separated, and with much finer resolution in theta, such that one can follow how the spectrum evolves as theta is varied. Or is it the point that the spectrum is never continuous? Also, do figs 1b and 1d show exactly the same information, just with different y-axis labels?

Reply: We would like to thank the reviewer for the kind suggestion. We have presented a taller figure with much finer resolutions in theta to clearly show the evolution of eigen-spectra. In addition, Fig. 1b shows the eigen-spectrum of two anyons in the 1D lattice model, and Fig. 1d gives the frequency-spectrum of the designed circuit simulator. Due to the nonlinear relationship ($f = f_0/(\epsilon + 4 + C_e/C)^{1/2}$) between the eigenfrequency of the circuit simulator (f) and the eigenenergy of two anyons (ϵ), the distribution of the eigen-spectrum for the circuit simulator should deviate from that of two anyons. In this case, by setting a relatively large grounding capacitor C_e at each circuit node, the circuit eigenfrequencies could show the near-perfect linear relationship with two anyon eigenenergies (equally spaced eigen-frequencies), which is crucial for the realization of anyonic BOs in circuit simulators. Detailed numerical results of frequency-spectra with different C_e are shown in Fig. S1.

Action taken:

- In the revised manuscript, we have changed Figs. 1b and 1d to taller figures with much finer resolutions in theta.

6). How are the curves in Figs 3b, c related to the oscillations in figs 2b,c,e,f? They certainly look quite different.

Reply: Previous results in Fig. 3 are the envelope curves of the measured voltage signals at the circuit node (6, 5). For a clear comparison between the numerical and experimental results, we present the calculated and measured voltage signals in the same form of another circuit node (5, 5) with a better oscillation behavior, as shown in Figs. 2f, 2j and Figs. 3f, 3i. Numerical and experimental results are consistent with each other, where the damped BO in experiments is due to the lossy effect in real circuit samples (see S7 of Supplementary Materials for the numerical results with different losses).

7). Some care may be needed to distinguish "numerical simulations" and "circuit simulations". Perhaps the experimental results/setup can be referred to as "experiments" so as to not confuse them with the (rigorously done) numerical simulations of the circuits?

Reply: Following the reviewer's suggestion, the experimental results/setup are referred to as "experiments" in the revised manuscript.

8). In the supplement section on two bosons and two pseudofermions, INICs are used. Such INIC networks have been implemented in PRL 126 (21), 215302, even if the eigenvalues are still real.

Reply: We would like to thank the reviewer for the kind recommendation of the related paper. We have cited it as Ref. [61] in the revised manuscript.

9). Is the circuit still Hermitian? And are the different oscillation frequencies then remotely related to non-Hermitian mechanisms, for instance in doi.org/10.1038/s42005-020-00417-y?

Reply: We would like to thank the reviewer for the kind recommendation of the related paper. We have cited it as Ref. [69] in the revised manuscript. Actually, with the existence of parasitic resistances, our fabricated circuit samples are indeed non-Hermitian systems. To clarify the influence of losses on anyonic BOs, we extend the original two-anyon lattice model to contain the intrinsic dissipation rate. In this case, the original two-anyon model can be rewritten as:

$$H = -J \sum_{l=1}^N (a_l^\dagger a_{l+1} + a_{l+1}^\dagger a_l) + i \sum_{l=1}^N \gamma n_l + F \sum_{l=1}^N l n_l.$$

Based on such a non-Hermitian model, we give numerical results of BOs for two bosons and two pseudofermions. Here, we set $\gamma=0.1$, and other parameters are the same as those used in Fig. S3 of the Supplementary Materials. As shown in Figs. R4a and R4b, we calculate the evolution of $|c_{mn}(t)|$ with $\theta = 0$ and $\theta = \pi$. Moreover, Figs. R4c and R4d display the evolution of a fixed state $|c_{5,5}(t)|$ with $\theta = 0$ and $\theta = \pi$, respectively. It is clearly shown that damped periodic dynamics of both bosons and pseudofermions appear, and the oscillation period of the two bosons is still twice of that for two pseudofermions. Such damped BOs are consistent with the measured result in circuits. In this case, losses in the fabricated circuit could be mapped to the dissipation rate of the 1D anyonic lattice model.

Fig. R4. (a) and (b) The evolution of two-anyon state $|c_{mn}(t)|$ under the intrinsic dissipation rate with $\theta = 0$ and $\theta = \pi$. (c) and (d) The evolution of a fixed state $|c_{5,5}(t)|$ with $\theta = 0$ and $\theta = \pi$.

Action taken:

- In the revised manuscript, we have added the following discussion to page 12 to illustrate the influence of the dissipation rate on the BOs: *“It is worthy to note that the loss in fabricated circuits could be mapped to the dissipation rate of the 1D anyonic lattice model. To clarify the influence of losses on the anyonic BOs, we extend the two-anyon lattice model in Eq. (1) to contain the intrinsic dissipation rate [67] (see S8 of the Supplementary Materials for details). In this case, similar to the measured voltage dynamics, the damped BOs of two anyons also appear.”*
- In the S8 of revised Supplementary Materials, we have added numerical results of damped BOs of two bosons and two pseudospins based on the 1D lattice model with dissipations.

10). Last but not least, do check the manuscript for grammatical and spelling errors i.e. "still a great challenge" instead of "still a great challenging" in line 58, also the sentence seems incomplete, starting from "While". In line 59, are newly accessible platforms really "expected" ? In line 271, "except" makes the sentence sound awkward.

Reply: We would like to thank the reviewer for the kind suggestion. We have carefully checked and modified the grammatical and spelling errors in the manuscript.

Response to comments of reviewer #2

This work studies a model for a pair of particles in which the statistical phase can be varied. The authors demonstrate that the two-particle 1D model can be mapped onto a single-particle 2D lattice. They then demonstrate that this model can be simulated with a suitably-designed circuit. Finally, the authors fabricate the circuit for the cases of bosons and pseudo-fermions and measure voltage oscillations. They observe oscillations in the envelope function matching their simulations and theoretical predictions, providing validation for their approach to simulation of the quantum model.

This is an interesting work which illustrates the power of electronic circuits to model exotic quantum phenomena which are difficult to realize experimentally. The novelty of this work is enhanced by the experimental implementation of the circuit. Thus, I believe this work should be published. However, I have a few questions and suggestions for the authors

Reply: We would like to thank the reviewer for the positive evaluation and valuable suggestions of our work. In the following, we will give a detailed response to all points proposed by the reviewer.

1). The authors should emphasize that the theoretical model they describe starting on line 72 was developed by Longhi and Valle, PRB 85, 165144 (ref. 42).

Reply: We would like to thank the reviewer for the comment. Following the reviewer's suggestion, we have further emphasized that the theoretical model described in our work was developed by Longhi and Valle, PRB 85, 165144 (ref. 45).

Action taken:

- In the revised manuscript, we have added the following discussion on page 3 to emphasize that the theoretical model described in our work was developed by Longhi and Valle, PRB 85, 165144 (ref. 45):
“*Following the theoretical model proposed by Longhi and Valle [45], we start by considering a pair of non-interacting anyons hopping on a one-dimensional (1D) chain subjected to an external force F .*”.

2). There are additional theoretical works which provide the basis for the 1D anyon model used in this work. I believe it is appropriate for the authors to cite A. Kundu, PRL 83, 1275–1278 and T. Keilman et al, Nat. Comm. 2, 361 which develop the framework for 1D anyons and pseudofermions.

Reply: We would like to thank the reviewer for the kind recommendation of the related paper. We have cited these two papers as Ref. [34] and Ref. [35] in the revised manuscript.

3). Can the authors describe in more detail how the excitation is applied to the simulated and real circuits? Is a short pulse used, or is it a continuous excitation starting from $t=0$? Additionally, can the authors discuss what this circuit excitation corresponds to physically in the anyon-Hubbard model?

Reply: We would like to thank the reviewer for the comment. The continuous voltage signals at a fixed frequency are injected into the designed electric circuit for simulations and measurements. In particular, the voltage pseudospin in the two-pseudofermion circuit simulator is suitably excited by setting the input signal as $V_{(12,12),1} = V_0 e^{i2\pi f t}$, $V_{(12,12),2} = -V_0 e^{i2\pi f t}$. It is worthy to note that between the excited circuit node and the input source, we connect a suitable capacitor to ensure that the periodic oscillation

could also appear at the excited node (12,12). Such a circuit excitation corresponds to setting the probability amplitude of the input two-anyon state as $\psi_{\text{in}}(t) = \delta_{m,12}\delta_{n,12}c_{mn}e^{i\epsilon t}$ in the anyonic lattice model. The corresponding numerical results of two-anyon BOs with such an initial excitation (shown in Fig. S3 of the Supplementary Material) are consistent with that of the designed electric circuits.

Action taken:

- In the revised manuscript, we have added the following discussion in page 7 to illustrate the excitation detail in the two-boson simulator: “*Here, the excitation frequency is set as $f=1.56$ MHz, and the central circuit node is excited by $V_{(12,12),1} = V_0e^{i2\pi ft}$. It is worthy to note that between the excited circuit node and the input source, we connect a suitable capacitor to ensure that the periodic oscillation could also appear at the excited node.*”
- In the revised manuscript, we have added the following discussion in page 8 to illustrate the excitation detail in the two-pseudofermions simulator: “*The voltage pseudospin can be suitably excited by setting the input signal as $V_{(12,12),1} = V_0e^{i2\pi ft}, V_{(12,12),2} = -V_0e^{i2\pi ft}$.*”

4). It is found that in the physical circuit, the Bloch oscillations decay with time due to losses. Do these losses correspond to any physical processes in the anyon-Hubbard model, i.e. leakage of particles out of the system?

Reply: We would like to thank the reviewer for the comment. Losses in the circuit could be mapped to the dissipation rate of the anyonic lattice model. To clarify the influence of losses, we extend the original two-anyon lattice model to contain the intrinsic dissipation rate. In this case, the original two-anyon model can be rewritten as:

$$H = -J \sum_{l=1}^N (a_l^\dagger a_{l+1} + a_{l+1}^\dagger a_l) + i \sum_{l=1}^N \gamma n_l + F \sum_{l=1}^N l n_l.$$

Based on such a non-Hermitian model, we give numerical results of BOs for two bosons and two pseudofermions. Here, we set $\gamma=0.1$, and other parameters are the same as those used in Fig. S3 of the Supplementary Materials.

Fig. R5. (a) and (b) The evolution of two-anyon state $|c_{mn}(t)\rangle$ under the intrinsic dissipation rate with $\theta = 0$ and $\theta = \pi$. (c) and (d) The evolution of a fixed state $|c_{5,5}(t)\rangle$ with $\theta = 0$ and $\theta = \pi$.

As shown in Figs. R5a and R5b, we calculate the evolution of $|c_{mn}(t)|$ with $\theta = 0$ and $\theta = \pi$. Moreover, Figs. R5c and R5d display the evolution of a fixed state $|c_{5,5}(t)|$ with $\theta = 0$ and $\theta = \pi$, respectively. It is clearly shown that damped periodic dynamics of both bosons and pseudofermions appear, and the oscillation period of the two bosons is still twice of that for two pseudofermions. This phenomenon is consistent with the observed damped BOs in circuits.

Action taken:

- In the revised manuscript, we have added the following discussion in page 12 to illustrate the influence of the dissipation rate on the BOs: *“It is worthy to note that the loss in fabricated circuits could be mapped to the dissipation rate in the 1D anyonic lattice model. To clarify the influence of losses on the anyonic BOs, we extend the two-anyon lattice model in Eq. (1) to contain the intrinsic dissipation rate [67] (see S8 of the Supplementary Materials for details). In this case, similar to the measured voltage dynamics, the damped BOs of two anyons also appear.”*
- In the S8 of revised Supplementary Materials, we have added numerical results of damped BOs based on the 1D anyonic lattice model.

5). In the Supplementary Material S1, there is a parameter C_U , which I did not find mentioned elsewhere in the manuscript. What is this capacitance?

Reply: We would like to thank the reviewer for the comment. The capacitor C_U (grounding at the circuit node on the diagonal) corresponds to the onsite interaction (U) between two anyons. In the main text, we set the onsite interaction between two anyons to zero. Because, the existence of onsite interactions could suppress the statistical angle dominated BOs [45]. Due to the fact that the onsite Hubbard interaction (U) is not used in the manuscript, to avoid confusing, we have deleted the onsite interaction term (U) in Eq. (1).

6). I am not sure that it makes sense to describe the simulated particles as "anyons." Anyons are generally understood to be distinct from bosons and fermions in having a statistical phase different from 0 or π . While the model that the authors employ can be generalized to describe anyons, their simulations and realized circuits have focused on the statistical phases of 0 and π , corresponding to bosons and pseudofermions rather than anyons. Thus, a more accurate title might be "Observation of Bloch oscillations for bosons and pseudo-fermions."

Reply: We would like to thank the reviewer for the kind suggestion. We have changed the title to "Observation of Bloch oscillations dominated by particle statistics".

Response to comments of reviewer #3

In the manuscript “Observation of Anyonic Bloch oscillations” the authors Zhang et al. discuss a realization of two-particle anyon physics by coupled electric circuits and they observe the effect of statistics through the analysis of Bloch oscillations.

In my opinion, the manuscript has an interesting aim but the actual content is quite modest. In particular, the experimental results are limited to a single plot in Fig. 3 and a single quantity, the amplitude V on a certain element of the circuitry, whereas most of the theoretical ideas are just a repetition of Ref. 42 (Longhi, Della Valle PRB 2012), which are then applied to the circuit platform. Besides, the anyonic Bloch oscillations for $\theta=\pi$ are then relegated to the last figure panel of the manuscript, Fig.3c, and represented by a very small and barely observable peak, which should have been the main highlight of the paper but rather seems a more modest outcome. As an example of more detailed analysis, e.g. for Fig. 3, it would have been useful to quantitatively estimate the losses and show that the measured damping can be explained by a more refined model, if that is possible. Besides, is there no other measurement to show in order to validate or present the results except for the dynamics of a single site? Is there no other measurement beyond the ones already discussed by Longhi et al.? Moreover, I may disagree with the authors' statement (which contains no further motivation) that the anyonic Bloch oscillations are an intractable challenge for quantum/classical platforms as a motivation for their work. I will comment further on this here below.

To summarize: I think that the paper does not contain 1) a significant amount of experimental results 2) a significant amount of innovative content that justifies publication in Nature Communications. I may change my mind with a substantial improvement of the experimental results and of the theoretical modeling, namely, with major improvements of the manuscript and a more in-depth analysis of the system and the physics that they can access. Here below some further points that the author may consider to address.

Reply: We would like to thank the reviewer for the careful review and valuable suggestions, which help to greatly improve the manuscript. Firstly, we appreciate the reviewer for agreeing that our manuscript has an interesting aim. On the other hand, we also note that our previous manuscript does not contain a significant amount of experimental results. Following kind suggestions of the reviewer, in the revised manuscript, we have added more detailed experimental results to illustrate the anyonic BO (see following comments for details). Moreover, the lossy and disorder effects are theoretically investigated, giving a more accurate model to explain the circuit measurements. Additionally, we also numerically demonstrate that the Wannier Stark spectrum of two anyons could also appear at other statistic angles beyond bosons and pseudofermions under a suitable value for the ratio of the external forcing to the hopping rate (F/J). The corresponding discussions are added in the conclusion part to illustrate the exotic anyonic BO that has not been revealed previously. To summarize, the improvement of our manuscript lies in the following:

- In Fig. 2 and Fig. 3 of the revised manuscript, we have added simulated and measured results of the frequency-dependent impedance response of two-boson and two-pseudofermion circuit simulators to illustrate the existence of the Wannier-Stark ladder in fabricated circuit simulators.

- In Fig. 2 and Fig. 3 of the revised manuscript, we have added simulated and measured results of 2D voltage distributions in two-boson and two-pseudofermion circuit simulators at different times with equal intervals (within a period) to illustrate that the required parities of two-boson and two-pseudofermion are always conserved in BOs.
- In S6 of the revised Supplementary Materials, we have added numerical results of BOs based on the anyonic lattice model with dissipation rates. The damped BOs induced by dissipations are consistent with that observed in circuits.
- In S7 of the revised Supplementary Materials, we have calculated the voltage dynamics in designed circuits with different losses. Compared to experimental results, we can deduce that the effective series resistance of inductance in the fabricated circuit is approximately $50m\Omega$.
- In S8 of the revised Supplementary Materials, the influence of circuit disorders on the BOs and asymmetric wavefunction of pseudofermions are numerically investigated.
- We deduce that with a balance between the quantum statistic induced splitting of Wannier Stark ladder and energy-level coupling which is determined by the external force and the statistic angle, the newly form Wannier Stark spectrum with equal energy-spacings could appear at a suitable statistic angle θ beyond bosons and pseudofermions. For example, when $F/J=1.195$ and $\theta = 1.16$, we numerically demonstrate that the Wannier Stark spectrum of two anyons appears, and the period of the corresponding BO is three times of that for two bosons in S10 of the revised Supplementary Materials.

In the following, we give detailed responses to each review's comment and hope that our efforts can convince the reviewer for the recommendation of publishing.

1). The authors state that the realization of the effect is an intractable challenge for quantum/classical platforms without providing further motivation. Unless they provide a striking argument, I will strongly disagree with it. As shown in Fig. 1a the lattice can be realized by tailoring certain hopping coefficients near the main diagonal, and in particular by realizing a negative hopping to obtain pi-flux on certain plaquettes for the $\theta=\pi$ case. This can easily be realized in photonics (see for example ring resonators Mittal et al. Nature Photonics volume 13, pages692–696 (2019) or optical waveguides Mukherjee et al. Phys. Rev. Lett. 121, 075502 (2018), Kremer et al. Nature Communications volume 11, Article number: 907 (2020)).

Reply: We would like to thank the reviewer for the comment. It is true that our original statement is not accurate. We have changed it to “*Although the statistic-induced halving of the BO frequency is a very interesting phenomenon, to date, the experimental observation of such an exotic effect is still lacking even using the potential artificial structures [46-48]*”. In addition, we have cited these papers as Refs. [46-48] in the revised manuscript.

2). The authors write an anyon Hubbard model in Eq. 1 but the onsite U Hubbard interaction is never used in the manuscript. I do not understand why they are therefore talking about an interacting model, when interactions do not play a role in the analysis presented here (but they are used in Ref. 42).

Reply: We would like to thank the reviewer for the comment. It is true that we have not considered the onsite interaction between two anyons. The existence of onsite interaction could suppress BOs dominated by the quantum statistics. To avoid confusing, in the revised manuscript, we have deleted the onsite interaction term (U) in Eq. (1).

3a). The commutation relations in Eq. 2 show that the particles behave as fermions when they are at different sites. This translates, for $\theta=\pi$, to $c_{nm}=-c_{mn}$. Is this condition fulfilled in their results? In principle this is an extra constraint which is not necessarily realized away from the diagonal, unless the modes are projected to the antisymmetric sector. Without this condition in place, the claim that the experiment is observing anyonic statistics for two particles is not true. If one looks at the spectrum, it should contain also states where the antisymmetric constraint is not fulfilled in a certain range. So, the question can be rephrased by clarifying if the "wavefunction" is in the right sector when they observe the time dynamics, namely if the corresponding parity is conserved over time. The authors should also estimate how imperfections of the setup may lead to a deviation from this. This could be a nice piece of information that makes the analysis less straightforward, for example.

Reply: This is an important point raised by the reviewer. To clearly illustrate the anyonic properties, distributions of voltage amplitude and phase are calculated for both circuit simulators at different times with equal intervals (in a period), as shown in Fig. R6, where sub-plots from left to right correspond to results with increased times. Here, the voltage amplitudes are normalized by the maximum voltage signal. We can see that the absolute value of voltage at circuit nodes (m, n) and (n, m) are identical for both two-boson and two-pseudofermion circuit simulators. As for the phase distribution, it is shown that phases at circuit nodes (m, n) and (n, m) are always the same in the two-boson simulator, which is matched to the commutation relation of two bosons $c_{mn} = c_{nm}$. While, the phase difference between circuit nodes (m, n) and (n, m) in the two-pseudofermion simulator always equals to π (except for the initial time), that is also consistent with the requirement of the commutation relation for two pseudofermions $c_{mn} = -c_{nm}$. These results clearly show that the required parities of two-boson and two-pseudofermion are always conserved in the BOs. The associated experimental results are presented in Fig. R7. We can see that the phase distributions are still consistent with the corresponding commutation relations. The little derivation of measurements results from the weak disorder effects in the fabricated circuit simulator (as discussed below). Moreover, the measured voltage amplitudes show damped BOs due to the lossy effect.

Then, following the reviewer's suggestion, we have also numerically investigated the influence of disorder effects on the voltage phase distribution and BOs in two-boson and two-pseudofermion circuit simulators. As shown in Fig. R8a and Fig. R8c, we calculate the voltage dynamics of all nodes in the designed two-boson and two-pseudofermion simulators with fluctuations of circuit elements being 0.5%, 1%, 3% and 5%, respectively. The corresponding single-node amplitudes at (5,5) are plotted in Figs. R8b and R8d, respectively. We can see that the period of BOs is still maintained with the small disorder strength. While, when the disorder strength approaches to 3%, the BOs are destroyed for both two-boson and two-pseudofermion circuit simulators. Moreover, the phase distributions of voltage signal at a fixed time (the half in the first BO period) but different disorder strengths are displayed in Fig. R9 with the fluctuations of circuit elements being set as 0.5%, 1%, 3% and 5%. In this case, it is shown that the symmetric and antisymmetric distributions of the voltage signal in two-boson and two-pseudofermion circuit simulators are significantly broken when the disorder strength reaches to 3%, being consistent with the BO behavior in Fig. R8. In our sample, to ensure the tolerance of circuit

elements to be as low as possible, we use a WK6500B impedance analyzer to select circuit elements with a high accuracy (the averaged disorder strength is less than 1%).

Fig. R6. (a) and (b) Numerical results for the distribution of voltage amplitude and phase at different times (in a period) for two-boson and two-pseudofermion circuit simulators.

Fig. R7. (a) and (b) Experimental results for the distribution of voltage amplitude and phase at different times (in a period) for two-boson and two-pseudofermion circuit simulators.

Fig. R8. (a) and (c) The voltage dynamics of all nodes in the two-boson and two-pseudofermion circuit simulators with fluctuations of circuit elements being 0.5%, 1%, 3% and 5%. (b) and (d) The voltage amplitude at the (5,5) node in the two-boson and two-pseudofermion circuit simulators with fluctuations of circuit elements being 0.5%, 1%, 3% and 5%.

Fig. R9. The phase distributions of voltage signal at a fixed time (the half in the first BO period) under different disorder strengths 0.5%, 1%, 3% and 5% for the two-boson (a) and two-pseudofermion (b) circuit simulators.

Action taken:

- In the revised manuscript, we have added simulated and measured results for the 2D voltage distributions at different times in Fig. 2 and Fig. 3.
- In the S6 of revised Supplementary Materials, we have added simulation results about the influence of disorder effects on the voltage phase distribution and BOs in two-boson and two-pseudofermion circuit simulators.
- In the revised manuscript, we have added following discussions in page 7 to illustrate the simulated 2D voltage distributions for two bosons: “*Fig. 2c displays the time-dependent evolution of $|V_{[m,n],1}(t)|^2$ at all nodes, where the circuit node (m, n) is labeled by $(m-1)N+n$. Amplitude (normalized to the maximum) and phase distributions of voltage signals at selected times with equal intervals (in the first period) are presented in Figs. 2d and 2e. We can see that voltages at symmetric circuit nodes (m, n) and (n, m) are always the same, being consistent with the commutation relation of two bosons.*”.
- In the revised manuscript, we have added following discussions in page 8 to illustrate the simulated 2D voltage distributions for two pseudofermions: “*Figs. 2h and 2i show normalized amplitude and phase distributions of the voltage pseudospin at different times in the first period. We can see that the phase difference between a pair of circuit nodes located at (m, n) and (n, m) always equals to π (except for the initial time), that is consistent with the requirement of anyonic commutation relation for pseudofermions.*”.
- In the revised manuscript, we have added following discussions in page 11 to illustrate the measured 2D voltage distributions for two bosons: “*The measured amplitude and phase distributions of voltage signal at different times (in the first period) are plotted in Fig. 3d. We can see that the symmetric voltage distribution exists. The little derivation should result from the weak disorder effects in the circuit (see S6 of the Supplementary Materials).*”.
- In the revised manuscript, we have added the following discussion in page 11 to illustrate the measured 2D voltage distributions for two pseudofermions: “*As shown in Fig. 3f, the measured amplitude and phase distributions of $V_{\perp,[m,n]}(t)$ at different times in the first period are presented. It is shown that the nearly asymmetric phase distribution is observed. Similar to the case of two bosons, the little derivation should result from the disorder effects in the circuit sample.*”.

3b). The commutation relations of interest to the authors have been discussed and brought to the audience of synthetic systems by Keilmann et al. Nature Communications volume 2, Article number: 361 (2011), which cites other papers on this topic as well. I suggest the authors to review the reference list accordingly besides citing just Ref. 42.

Reply: We would like to thank the reviewer for the kind recommendation of the related paper. We have cited the following papers in the revised manuscript as Refs. [34-36].

[34]. Kundu, A., Exact solution of double δ function Bose gas through an interacting anyon gas, *Phys. Rev. Lett.* **83**, 1275 (1999).

[35]. Keilmann, T., Lanzmich, S., McCulloch, I. et al. Statistically induced phase transitions and anyons in 1D optical lattices. *Nat Commun* **2**, 361 (2011).

[36]. Batchelor, M. T., Guan, X.-W. & Oelkers, N. One-dimensional interacting anyon gas: low-energy properties and haldane exclusion statistics. *Phys. Rev. Lett.* **96**, 210402 (2006).

4). In Fig. 1b-d, what is the reason for plotting the spectrum in that specific range? This is not clear to me and should be motivated in the text.

Reply: We would like to thank the reviewer for the comment. The frequency range presented in Fig. 1d corresponds to the energy range $(6F, 24F)$ of two-boson Wannier-Stark ladder, that is within the energy range $(2F, 46F)$ of the Wannier-Stark ladder for two bosons in the lattice with length being $N=23$. For a clear illustration of equally spaced eigen-spectra for two bosons and two pseudofermions, only a part of the eigen-spectra in the range of $(6F, 24F)$ is presented. In addition, the reason for plotting the eigen-spectrum in this specific range is because the excitation frequency of circuit simulators is set as 1.56 MHz in simulations and measurements, which is within the frequency range that we plotted in Fig. 1d.

Action taken:

- In the revised manuscript, we have added the following discussion in page 6 to clarify the reason for plotting the spectrum in that specific range: “*As shown in Fig. 1d, the frequency-spectrum related to eigen-energies in the range of $(6F, 24F)$ is plotted, where the excitation frequency (1.56MHz) used in simulations and measurements (discussed below) is located within this frequency range.*”.

5). I have not found an explanation of the fact that $\theta=\pi$ anyons have a doubling of Bloch oscillations period. From what I have seen in the manuscript, the authors limit themselves to state that this is the result in Ref. 42. I think this can be clarified and further developed. In general, the Wannier Stark spectrum and gaps can be deduced for a single band and the authors could elaborate further in this direction.

Reply: We would like to thank the reviewer for the comment. It is known that the Wannier Stark ladder possessing equally spaced eigen-energies is the key origin for the appearance of BOs. For two non-interacting BOs, the highly degenerated Wannier Stark spectrum exists. While, by introducing the anyonic statistic correlation, the degeneration of Wannier Stark spectrum of two bosons could be destroyed, making the original equally spaced energy-spectrum become irregular. The interesting thing disclosed by the previous work [45] is that if the ratio of the external forcing to the hopping rate (F/J) become smaller than 0.5, the near-perfect Wannier Stark spectrum could appear for pseudofermions (green dots), as shown in Fig. R10a with $F/J=0.5$. When the ratio of F/J gets increased, there is no equally spaced eigen-spectrum exists for pseudofermions (green dots), as shown in Fig. R10b with $F/J=1.195$. Here, the energy spectra are plotted within the range of $(6F, 24F)$.

Comparing Fig. R10a with Fig. R10b, we can see that the statistic induced energy-level coupling between different anyonic bands evolved from the original Wannier Stark spectrum (at $\theta = 0$) is dependent on the value of F/J . The smaller the F/J is, the larger energy-level coupling could be produced by the quantum statistic. In this case, we can see that the newly formed Wannier Stark spectrum of pseudofermions could be regarded as the rearrangement of the original Wannier Stark spectrum (at $\theta = 0$) assisted by the energy coupling induced by the statistic correlation.

Fig. R10. (a) and (b) Calculated eigen-energies of two anyons as a function of the statistical angle θ with $F/J=0.5$ and $F/J=1.195$, respectively.

It is worthy to stress that, beyond the conclusion given by Longhi and Valle (PRB 85, 165144), we further find the near-perfect Wannier Stark spectrum could also appear at other statistic angle under a suitable value of F/J . For example, as shown by black dots in Fig. R10b, a near-perfect Wannier Stark spectrum could also appear at $\theta = 1.16$ (black dots) with $F/J=1.195$, where the energy spacing is 1/3 of that for two bosons (blue dots), making the BO period of two anyons with $\theta = 1.16$ become three times of that for two noninteracting bosons. To illustrate such a novel anyonic BOs, we calculate the evolution of the probability amplitude of two anyons with $\theta = 1.16$, as shown in Figs. R11a and R11b. Here, the lattice size and external force are set as $N=15$ and $F/J=1.195$, and the input two-anyon state is $\psi_{\text{in}}(t) = \delta_{m,8}\delta_{n,8}c_{mn}e^{i20t}$. For comparison, the evolution results of two bosons are presented in Figs. R11c and R11d. It is clearly shown that the period of the anyonic BO with $\theta = 1.16$ is three times of that for two bosons, being consistent with the calculated Wannier Stark spectra.

From the above results, we deduce that with a balance between the quantum statistic induced splitting of Wannier Stark ladder and energy-level coupling which is determined by the external force and the statistic angle, the newly form Wannier Stark spectrum with equal energy-spacings may appear at a suitable statistic angle θ beyond bosons and pseudofermions. Analytically finding the requirement (the relationship between θ and F/J) for the existence of two-anyon Wannier Stark spectra, and simulating anyonic BOs with statistic angles beyond $\theta = 0, \pi$ in circuit networks (or other platforms) are interesting, and we remain these problems in future works.

Fig. R11. (a) and (c) The calculated evolution of the probability amplitude of two anyons with $\theta = 1.16$ and $\theta = 0$. (b) and (d) The evolution results of $|c_{8,8}(t)|$ with $\theta = 1.16$ and $\theta = 0$. Here, the lattice size and external force are set as $N=15$ and $F/J=1.195$, and the input two-anyon state is $c_{8,8}(t) = e^{i2\theta t}$.

Action taken:

- In the revised manuscript, we have added the following discussions in page 12: *“In addition, it is worthy to stress that the near-perfect Wannier Stark spectrum could also appear at other statistic angles (besides $\theta = 0$ and $\theta = \pi$) under a suitable value for the ratio of the external forcing to the hopping rate (F/J). For example, when we set $F/J=1.195$, the nearly equal-spaced energy-spectrum exists for two anyons with $\theta = 1.16$, where the energy spacing is 1/3 of that for two bosons, making the period of the anyonic BO with $\theta = 1.16$ become three times of that for two bosons (see S10 of the Supplementary Materials for details). In this case, we deduce that with a balance between the quantum statistic induced splitting of Wannier Stark ladder for two bosons and the coupling of anyonic bands evolved from different energy-level of the Wannier Stark spectrum at $\theta = 0$, the equally spaced two-anyon energy-spectrum could appear with suitable values of θ and F/J . Such a particle statistics dominated BO beyond bosons and pseudofermions is firstly revealed, and it could also be simulated by designed RLC circuit networks combined with a negative impedance converter with current inversion [68].”*
- In the S10 of revised Supplementary Materials, we have added numerical results of anyonic BOs with $\theta = 1.16$ and $F/J=1.195$.

6). The role of dissipation is not quantified but only mentioned. If this had a more quantitative analysis, it could explain the experimental results through a more refined theoretical modeling. An example of spectral analysis in the dissipative case is for instance provided by the work “Gorlach et al. Phys. Rev. A 98, 063625 (2018)” for the simulation of two-particle physics. If the single-site damping is understood, the modeling of dissipation could be used to quantify the observation in time displayed in Fig. 3. This analysis would show if dissipation is or is not the only mechanism in place for the observed outcome, perhaps disorder plays also a role. All these details should be explored and clarified a bit more in depth for an experimental paper.

Reply: We would like to thank the reviewer for the comment. To clarify the influence of losses on BOs, we extend the original two-anyon lattice model to contain an intrinsic dissipation rate. In this case, the original two-anyon lattice model can be expressed as:

$$H = -J \sum_{l=1}^N (a_l^\dagger a_{l+1} + a_{l+1}^\dagger a_l) + i \sum_{l=1}^N \gamma n_l + F \sum_{l=1}^N l n_l.$$

where γ is the dissipation rate. Based on such a non-Hermitian model, we give numerical results of BOs for two bosons and two pseudofermions. Here, we set $\gamma=0.1$, and the other parameters are the same as those used in Fig. S3 of the Supplementary Materials. As shown in Figs. R12a and R12b, we calculate the evolution of $|c_{mn}(t)|$ with $\theta = 0$ and $\theta = \pi$. Moreover, Figs. R12c and R12d display the evolution of a fixed state $|c_{5,5}(t)|$ with $\theta = 0$ and $\theta = \pi$, respectively. It is clearly shown that damped periodic dynamics of both bosons and pseudofermions clearly appear, and the oscillation period of the two bosons is still twice of that for two pseudofermions. This phenomenon is consistent with the observed BOs in circuits.

Fig. R12. (a) and (b) The evolution of two-anyon state $|c_{mn}(t)|$ under the intrinsic dissipation rate with $\theta = 0$ and $\theta = \pi$. (c) and (d) The evolution of a fixed state $|c_{5,5}(t)|$ with $\theta = 0$ and $\theta = \pi$.

Based on the above results, we can see that the lossy effect in circuits can map to the dissipation rate of lattice models for two anyons. To quantitatively estimate the loss of our circuit samples, we

calculate the voltage dynamics of all nodes in the designed circuit with the effective series resistances of inductance being $10\text{ m}\Omega$, $20\text{ m}\Omega$, $50\text{ m}\Omega$, $70\text{ m}\Omega$ and $100\text{ m}\Omega$, as shown in Fig. R13a for the two-boson simulator and Fig. R13c for the two-pseudofermion simulator. The corresponding voltage amplitude at the circuit node (5,5) are plotted in Fig. R13b and R13d, respectively. It is shown that with the series resistances of inductance being increased, the oscillated amplitude is significantly damped. Compared to the experimental results (in Figs. 3f and 3i), we can deduce that the effective series resistance of inductance in the fabricated circuit sample is approximately $50\text{ m}\Omega$.

Fig. R13. (a) and (b) The voltage dynamics of all nodes in the two-boson and two-pseudofermion circuit simulators with different losses. (c) and (d) The voltage amplitude at the (5,5) node in the two-boson and two-pseudofermion circuit simulators with different losses.

Furthermore, to clarify the influence of circuit disorder on the BOs, we further calculate the voltage dynamics of designed electric circuits with different fluctuations of circuit elements (0.5%, 1%, 3% and 5%), as shown in Fig. R8 (the above comment 3a). We can see that the BOs is still maintained under a relatively small disorder strength, where the BO period and amplitude are nearly unchanged. But, the BO is significantly destroyed with a large disorder strength. In our sample, to ensure the tolerance of circuit

elements to be as low as possible, we use a WK6500B impedance analyzer to select circuit elements with a high accuracy (the averaged disorder strength is less than 1%).

Action taken:

- In the revised manuscript, we have added the following discussion in page 12 to illustrate the influence of the dissipation rate on the BOs: *“It is worthy to note that the loss in fabricated circuits could be mapped to the dissipation rate in the 1D anyonic lattice model. To clarify the influence of losses on the anyonic BOs, we extend the two-anyon lattice model in Eq. (1) to contain the intrinsic dissipation rate [67] (see S8 of the Supplementary Materials for details). In this case, similar to the measured voltage dynamics, the damped BOs of two anyons also appear.”*
- In the S8 of revised Supplementary Materials, we have added numerical results of damped BOs based on the 1D anyonic lattice model.
- In the revised manuscript, we have added the following discussion in page 11 to illustrate the effective series resistances of inductances in the fabricated circuit: *“To fit the strength of loss in the fabricated circuit, we calculate the voltage dynamics with different series resistances of inductors (see S7 of the Supplementary Materials). In this case, we can deduce that the effective series resistance of inductance in the fabricated circuit is approximately 50mΩ.”*
- In the S6 and S7 of revised Supplementary Materials, we have added numerical results on the voltage dynamics of the designed electric circuit with different losses and disorders.

7). In line 207, the authors estimate the oscillation period (0.545ms). More details of this estimate (is it a fit, an extrapolation, a Fourier transform?) and of the corresponding error should be presented to the reader. Same comment applies to the experimental results.

Reply: We would like to thank the reviewer for noticing this point. The simulated oscillation period could be obtained by calculating the time difference between the first and second voltage maxima in the time domain of a circuit node. In particular, the circuit node (5,5) is used for the calculation of BO periods for bosons and pseudofermions. By simulating the voltage dynamics on many other representative circuit nodes and calculating the associated periods based on the same method, the period errors for two bosons and two pseudofermions are [-0.01ms, 0.012ms] and [-0.0068ms, 0.013ms], respectively. In experiments, the same method is used to obtain the BO period and associated errors. In particular, the measured errors of the two-boson and two-pseudofermion simulators are [-0.015ms, 0.017ms] and [-0.021ms, 0.016ms].

Action taken:

- In the revised manuscript, we have added the following discussion on page 7 to illustrate the method for obtaining the period and errors of two-boson BOs in simulations: *“The oscillation period could be obtained by calculating the time difference between two adjacent voltage maxima of a single circuit node. In this case, the associated oscillation period at the circuit node (5, 5) is approximately 0.548 ms, which is nearly consistent with the period $T_B = 1/\Delta f_B = 0.537ms$ predicted by the frequency-spectrum in Fig. 1d. By calculating the period on many other representative circuit nodes, the period error is in the range of [-0.01ms, 0.012ms]”*.
- In the revised manuscript, we have added the following discussion on page 11 to illustrate the period and errors of two-boson BOs in experiments: *“Based on the same method of obtaining the BO period in simulations, the measured damped BO period is approximately 0.535 ms (with the error being [-0.015ms, 0.017ms]), which is consistent with the simulation.”*

- In the revised manuscript, we have added the following discussion on page 9 to illustrate the period and errors of two-pseudofermion BOs in simulations: “*The calculated BOs period is approximately 1.08 ms (with the error being [-0.0068ms,0.013ms]), which is also consistent with $T_f = 1/\Delta f_f = 1.074ms$ predicted by the frequency-spectrum in Fig. 1d.*”.
- In the revised manuscript, we have added the following discussion on page 11 to illustrate the period and errors of two-pseudofermion BOs in experiments: “*We can see that the damped oscillation period is approximately 1.078 ms (with the error being [-0.021ms,0.016ms]), which is also consistent with the simulated result.*”.

8). I have also found a very unsatisfactory concluding paragraph, which does not provide much information about the impact of this work beyond the simple realization of two particle physics described by the Hamiltonian (1).

Reply: We would like to thank the reviewer for noticing this point. In the revised manuscript, we have added the following discussion to further illustrate the impact of our work: “*Our proposal could provide a flexible platform to further investigate and visualize many interesting phenomena related to particle statistics and other exotic few-particle physics. With the flexibility that the connection and grounding of circuit nodes are allowed in any desired way free from constraints of locality and dimensionality, the anyonic physics existing in the lattice model with nonlocal hopping and interactions (beyond nearest neighbors) could also be achieved. Moreover, by mapping the 1D multiple-anyon model to the higher-dimensional lattice model, the circuit network could also be used to simulate the anyonic physics with more particles. Furthermore, including nonreciprocal and non-Hermitian elements in the circuit network, the novel behavior induced by the interplay between the non-Hermitian effect [68] and the quantum correlation can be investigated. In addition, electric circuits are easily fabricated using existing chip manufacturing technology, making the achievable number of circuit nodes become extremely increased. Such an electric chip could make the designed circuit simulator implement much more complex anyonic physics, such as statistically induced phase transitions and statistics-related topological phases. Finally, the designed circuit simulator could also give a new way to manipulate the electronic signals with exotic behaviors.*”.

REVIEWER COMMENTS

Reviewer #1 (Remarks to the Author):

In their revision, the authors have significantly improved on the rigor and presentation of their experimental results, particularly on the 2D interpretation of their data. They have addressed all of my previous comments, and the revised manuscript is now a very solid and interesting experiment that will likely greatly excite the condensed matter and electrical engineering communities. As such, I would like to strongly recommend their work for publication in Nature Communications, with the following minor comments:

1. With the better presentation, it has now become clearer to me that the main idea behind the mapping from 2-particle system to a single-body 2d system, which the experiment is based on, has been also suggested in previous theoretical works, such as arXiv:2107.03414 and a few works cited therein. This does not dilute the value of this experimental manuscript, but it will be good to reference related previous works.

2. (Optional) I noted the title change, may I optionally also suggest "Observation of Bloch oscillations dominated by effective anyonic particle statistics".

Reviewer #2 (Remarks to the Author):

The authors have addressed my questions and concerns. I believe the revisions they have made are appropriate. Based on this, I recommend that the work should be published.

I do have one suggestion. The additional colormap plots showing voltage and phase in Fig. 2 and Fig. 3 make those figures very dense. I would suggest moving the colormap plots to separate figures to make the figures easier to read (i.e. have a new figure for the simulated colormap plots and an additional new figure for the experimental colormap plots).

Reviewer #3 (Remarks to the Author):

Dear editors,

I find the revised manuscript widely improved in content and now displaying a more adequate analysis. I am therefore recommending publication in Nature Communications.

I have however some minor comments, which I think should be clarified or addressed before final publication.

- In Fig. 3i, just before the first large peak ($t \sim 0.5$) there is another small peak ($t \sim 0.3$). It appears that also the numerical simulations in Fig.2j have a similar feature. Can the author understand and explain its origin?

- I have observed that certain sites (1-3) have a constant large occupation over the entire dynamics. This is explicitly evident in Fig.S12 for three sites in the center, but also appears in other figures, including Fig. 2c-d. I checked on Longhi's paper (e.g. Fig.2) and I do not find a similar feature. Could the author understand and explain it? I am just wondering whether it's an optical effect, a feature of the dynamics or a possible bug in the code.

- I am not completely satisfied by the justification of the period doubling for pseudo-fermions. The author just mention that this was shown in Longhi's paper, but I only find numerical evidence in that paper and not much physical insight concerning why this is happening, namely why the Wannier-Stark spectrum assumes that structure. It would have been useful to find an explanation for it, which the authors have still an opportunity to produce to make the results of their work more physically understandable.

- Similarly, the authors generalize the multiplication period of BOs to another statistical angle ($\theta = 1.16$) for $F = 1.195$. First question is: why this angle and this force? Is this a coincidence? How does the spectrum look like when it becomes dense (many sites)? It appears to be a coincidence, as the authors write that one needs to tune F/J to a precise value. Perhaps it could be that there is an underlying relation between F and θ , that provides Wannier-Stark spectra. In Longhi's paper there is a more stringent condition, namely that the force should be smaller than 0.5, which seems to hint at a deeper reason for this interference phenomenon to happen (which I asked the authors

to try to identify in my remark above and in the previous report, if possible). I may expect something similar to occur here with this fine-tuning, unless it's a coincidence. I find the present discussion therefore too vague and the authors should call the attention on these open questions if they want to keep such a paragraph in the text.

- While the text is generally well written and quite clear, I think that some improvement in the English presentation is still required. Despite the comments of one of the other Referees, I still identify this issue quite regularly across the text. To give an example: "little derivation" perhaps means "small deviation". Other unclear expressions or typos are also present and should be improved. The Supplementary Material text has more of these issues, e.g. in S10: "statistic angle"  "statistical angle", "statistics angle"  "statistical angle", "infeasible"  "unfeasible", "Because, the statistics" is a sentence that probably should attach to the previous one.

Draft Only

Response Letter to Reviewers

We are very grateful for the recommendation of publication on our revised manuscript (NCMMS-21-40151A) from all three reviewers.

In the text below, reviewer comments are quoted in blue and followed by our detailed response. We have also revised the manuscript and the Supplemental Materials based on the reviewer comments, and these updates are highlighted in red in those files. In the text below, these updates are also highlighted in *Italics*.

Response to comments of the reviewer #1

In their revision, the authors have significantly improved on the rigor and presentation of their experimental results, particularly on the 2D interpretation of their data. They have addressed all of my previous comments, and the revised manuscript is now a very solid and interesting experiment that will likely greatly excite the condensed matter and electrical engineering communities. As such, I would like to strongly recommend their work for publication in Nature Communications, with the following minor comments.

Reply: We would like to thank the reviewer for recommending the publication of revised manuscript.

1. With the better presentation, it has now become clearer to me that the main idea behind the mapping from 2-particle system to a single-body 2d system, which the experiment is based on, has been also suggested in previous theoretical works, such as arXiv:2107.03414 and a few works cited therein. This does not dilute the value of this experimental manuscript, but it will be good to reference related previous works.

Reply: We would like to thank the reviewer for the kind recommendation of the related paper. We have cited it as Ref. [49] in the revised manuscript and some related works are also cited as Refs. [29, 46-48].

2. (Optional) I noted the title change, may I optionally also suggest "Observation of Bloch oscillations dominated by effective anyonic particle statistics".

Reply: We would like to thank the reviewer for the kind suggestion. We have changed the title as "Observation of Bloch oscillations dominated by effective anyonic particle statistics".

Response to comments of reviewer #2

The authors have addressed my questions and concerns. I believe the revisions they have made are appropriate. Based on this, I recommend that the work should be published.

Reply: We would like to thank the reviewer for recommending the publication of revised manuscript.

I do have one suggestion. The additional colormap plots showing voltage and phase in Fig. 2 and Fig. 3 make those figures very dense. I would suggest moving the colormap plots to separate figures to make the figures easier to read (i.e. have a new figure for the simulated colormap plots and an additional new figure for the experimental colormap plots).

Reply: We would like to thank the reviewer for the kind suggestion. Due to the description order of each picture in Fig. 2 and Fig. 3, it is inconvenient to move colormap plots to separate figures. To make the figures easier to read, we re-design the arrangements and ratios of Fig. 2 and Fig. 3.

Draft Only

Response to comments of reviewer #3

Dear editors,

I find the revised manuscript widely improved in content and now displaying a more adequate analysis. I am therefore recommending publication in Nature Communications.

Reply: We would like to thank the reviewer for recommending the publication of revised manuscript.

I have however some minor comments, which I think should be clarified or addressed before final publication.

- In Fig. 3i, just before the first large peak ($t \sim 0.5$) there is another small peak ($t \sim 0.3$). It appears that also the numerical simulations in Fig. 2j have a similar feature. Can the author understand and explain its origin?

Reply: We would like to thank the reviewer for the comment. In a single period of the BO, the wave function of two anyons does not always monotonically increase or decrease. There are several mini peaks in each period of BOs beside the maximum peak, which result from the wave interference effect. And, the locations of these peaks in the time domain are also not fixed at different sites. To clarify this effect, as shown in Figs. R1a and R1b, we plot the time-dependent probability amplitude at lattice sites ($m=5, n=5$), ($m=8, n=8$) and ($m=9, n=9$) for two bosons and two pseudofermions, respectively. Other parameters are the same to those used in Fig. S3. We can see that each site may possess some mini peaks in a period of the BO, which clearly manifest the non-monotonical dynamics of each site. In this case, the non-monotonical dynamics of voltage signals in the circuit simulator is also consistent with the result of quantum lattice model at the (5, 5) site.

Fig. R1. (a) and (b) The time-dependent probability amplitudes at sites $(m=5, n=5)$, $(m=8, n=8)$ and $(m=9, n=9)$ for two bosons and two pseudofermions, respectively.

- I have observed that certain sites (1-3) have a constant large occupation over the entire dynamics. This is explicitly evident in Fig.S12 for three sites in the center, but also appears in other figures, including Fig. 2c-d. I checked on Longhi's paper (e.g. Fig.2) and I do not find a similar feature. Could the author understand and explain it? I am just wondering whether it's an optical effect, a feature of the dynamics or a possible bug in the code.

Reply: We would like to thank the reviewer for the comment. Actually, in Fig. S12 and other figures (such as Fig. S3 and Fig. 2), the occupation at the center site (the excited site) is not constant but also exhibits the periodic oscillation with large background occupations. To illustrate this behavior, the detailed dynamics at the center site (the excited site) of two bosons and two pseudofermions are calculated by solving the effective coupling equation (Eq. S13 and Eq. S14), as shown in Fig. S3c and S3d (also in Fig. R2a), where the probability amplitude of the input two-anyon state is set as $\psi_{in}(t) = \delta_{m,12}\delta_{n,12}e^{iet}$. It is clearly shown that the periodic BOs at the center site appear. As for the LC circuit simulator, the continuous voltage signals at a fixed frequency are injected into the designed electric circuit. We connect a suitable capacitor between the excited circuit node and the voltage source to ensure that the periodic oscillation could also appear at the excited node (12,12). Such a circuit excitation corresponds to setting the probability amplitude of the input two-anyon state as $\psi_{in}(t) = \delta_{m,12}\delta_{n,12}e^{iet}$. The calculated voltage dynamics at the excited node (12,12) for the two-boson and two-pseudofermion simulators are shown in Fig. R2b. We can see that the periodic BOs also appear at the excited sites in our designed circuit simulators.

In addition, the input state at the site (m, n) of Longhi's paper is in the form of $\psi_{in}(t) = \delta_{m,12}\delta_{n,12}$ without a fix driving energy. To precisely match the time-dependent Schrödinger equation of two pseudofermions and two bosons, we also designed another RC circuit simulator based on resistances and capacitances (see S9 in the Supplementary Materials) to precisely match the time-dependent Schrödinger equation of two pseudofermions and two bosons. In this case, the behavior of BOs in RC circuit simulators is the same to Longhi's results. Detailed numerical results are given in S9 of the Supplementary Materials.

Fig. R2. (a). The detailed dynamics of two bosons and two pseudofermions at the excited site with the probability amplitude of the input two-anyon state is set as $\psi_{in}(t) = \delta_{m,12}\delta_{n,12}e^{iet}$. (b). The calculated voltage dynamics at the excited node (12,12) for the two-boson and two-pseudofermion simulators.

In the revised manuscript, we have added the following discussion in page 7 to clarify the oscillated behavior at the center site: *“We connect a suitable capacitor between the excited circuit node*

and the voltage source to ensure that the periodic oscillation could also appear at the excited node (12,12). In this case, although the large occupation exists at the center site (the excited site), it is not a constant but also exhibits the periodic oscillation.”.

- I am not completely satisfied by the justification of the period doubling for pseudo-fermions. The author just mention that this was shown in Longhi's paper, but I only find numerical evidence in that paper and not much physical insight concerning why this is happening, namely why the Wannier-Stark spectrum assumes that structure. It would have been useful to find an explanation for it, which the authors have still an opportunity to produce to make the results of their work more physically understandable.

Reply: We would like to thank the reviewer for the comment. It is true that finding a physical explanation on the appearance of Wannier-Stark spectrum of pseudo-fermions is very useful to make the results more physically understandable. In general, the appearance of newly formed Wannier Stark ladder for two pseudo-fermions should result from a balance between the statistical angle (θ) and the ratio of external force to the hopping strength (F/J), which collectively control the splitting and coupling of anyonic energy bands. As for the case of two bosons, the mapped 2D lattice in Fig. 1a possesses a mirror symmetry with respect to the $m=n$ line, which protects the degeneration of Wannier-Stark spectrum. By introducing the statistical angle of two anyons, the mirror symmetry is broken, resulting in the splitting of the highly degenerated eigen-spectrum of two bosons. In this case, the eigen-spectra of two anyons are always not equal-spaced and the energy spacing is smaller than the bosonic counterpart. When the statistical angle reaches to $\theta = \pi$, many eigenmodes become nearly degenerated again. Under a relatively strong hopping condition ($F/J < 0.5$), the suitable energy-level coupling between different anyonic bands could make the Wannier Stark spectrum reappear for two pseudo-fermions. The lower spatial symmetry of mapped lattice model of pseudofermions compared to that of bosons leads to a smaller energy degeneracy and a denser distribution of eigen-energies. In this case, the energy spacing of pseudo-fermions is half of that for bosons, making the BO period of pseudo-fermions become doubling.

Action taken:

- In the revised manuscript, we have added the following discussion in page 5 to give a physical insight on the Wannier-Stark spectrum of pseudo-fermions: *“As for the case of two bosons, the mapped 2D lattice in Fig. 1a possesses a mirror symmetry with respect to the $m=n$ line, which protects the degeneration of different energy-levels (blue dots for $\theta = 0$), and the degenerated eigen-energies $\varepsilon_{mn} = (m + n)F$ are equally spaced in the form of the Wannier-Stark ladder with $\Delta\varepsilon = F$. By introducing the statistical angle of two anyons, the mirror symmetry is broken, resulting in a splitting of the highly degenerated eigen-spectrum of two bosons. In this case, the eigen-spectra of two anyons are always not equal-spaced and the energy spacing is smaller than the bosonic counterpart. When the statistical angle reaches to $\theta = \pi$, many eigenmodes become nearly degenerated again. Under a relatively strong hopping condition ($F/J < 0.5$), the suitable energy-level coupling between different anyonic bands could make the Wannier Stark spectrum reappear for pseudo-fermions. The lower spatial symmetry of mapped lattice model of pseudofermions compared to that of bosons leads to a smaller energy degeneracy and a denser distribution of eigen-spectrum. In this case, the energy spacing of pseudo-fermions is $\Delta\varepsilon = F/2$, which makes the BO frequency of two pseudofermions become half of that for two noninteracting bosons.”.*

- Similarly, the authors generalize the multiplication period of BOs to another statistical angle ($\theta=1.16$) for $F=1.195$. First question is: why this angle and this force? Is this a coincidence? How does the spectrum look like when it becomes dense (many sites)? It appears to be a coincidence, as the authors write that one needs to tune F/J to a precise value. Perhaps it could be that there is an underlying relation between F and θ , that provides Wannier-Stark spectra. In Longhi's paper there is a more stringent condition, namely that the force should be smaller than 0.5, which seems to hint at a deeper reason for this interference phenomenon to happen (which I asked the authors to try to identify in my remark above and in the previous report, if possible). I may expect something similar to occur here with this fine-tuning, unless it's a coincidence. I find the present discussion therefore too vague and the authors should call the attention on these open questions if they want to keep such a paragraph in the text.

Reply: We would like to thank the reviewer for the comment. The above reply is focused on the statistical angle induced splitting and coupling of the two-boson Wannier-Stark ladder, which result from the reduced spatial symmetry of the mapped lattice model. Actually, the energy-level splitting and coupling also dependent on the value of F/J . Because, the energy-level splitting and coupling are induced by the complex hopping ($J e^{\pm i\theta}$) around the $m=n$ line. Hence, the larger value of F/J is, the smaller of the energy-level splitting and coupling exist. In this case, under a relatively large value of F/J , the unsuitable energy-level coupling between different anyonic bands makes the eigen-spectrum of pseudofermions become unequally spaced. While, to give the accurate relationship between F/J and θ for the appearance of Wannier Stark ladder of pseudo-fermions, the energy spectrum of two anyons in the noninteracting limit should be analytically proposed, that is indeed a tricky problem. Because our work is focused on the experimental observation of anyonic BOs, we remain such a theoretical problem in future works.

Beyond the anyonic BO of two pseudo-fermions with $F/J < 0.5$, we find that the near-perfect Wannier Stark spectrum could also appear at other statistical angles with $F/J > 0.5$. The appearance of Wannier Stark spectrum with $\theta=1.16$ and $F/J=1.195$ should result from a balance between the quantum statistic (θ) and the ratio of external force to the hopping strength (F/J). These two factors could collectively control the splitting and coupling of anyonic energy bands. In this case, the Wannier Stark spectrum may appear at suitable values of statistical angle and an external forcing. The much lower spatial symmetry of mapped lattice model of two anyons (beyond bosons and pseudofermions) leads to a smaller energy degeneracy and a denser distribution of eigen-spectrum. In this case, the energy spacing of two anyons with $\theta=1.16$ is 1/3 of that for bosons, making the corresponding BO period become three times of that for two bosons. In addition, the eigen-spectrum stays the same with more sites in the considered energy range, as shown in Fig. R3a with $N=31$. Moreover, we note that the appearance of multiplication period of BOs is not a coincidence. For example, if the external forcing is changed to $F/J=1$, the near-perfect Wannier-Stark spectrum could also appear with the statistical angle being around $\theta=2.09$, as shown in Fig. R3b. Similarly, to give the accurate relationship between F/J and θ for the appearance of Wannier Stark ladder with $F/J > 0.5$, the energy spectrum of two anyons in the noninteracting limit should be analytically proposed, and we remain this problem in future works.

Fig. R3. (a) and (b) Calculated eigen-energies of two anyons as a function of the statistical angle θ with $F/J=1.195$ and $F/J=1.0$, respectively. Here, the lattice size is $N=31$.

Action taken:

- In the revised manuscript, we have changed the discussion on the appearance of anyonic BOs in page 12 to directly give the conclusion of anyonic BOs beyond bosons and pseudofermions: *“In addition, it is worthy to stress that the near-perfect Wannier Stark spectrum could also appear at other statistical angles (besides $\theta = 0$ and $\theta = \pi$) under a suitable value for the ratio of the external forcing to the hopping rate (F/J), where the corresponding period of the BO could become three times of that for two bosons (see S10 of the Supplementary Materials for details). Such a BO dominated by particle statistics beyond bosons and pseudofermions could also be simulated by designed RLC circuit networks combined with a negative impedance converter with current inversion [72].”*
- In the revised Supplementary Materials, we have added the calculated eigen-spectrum with $F/J=1$ and also added some physical insights on the appearance of Wannier-Stark spectrum beyond bosons and pseudofermions.

- While the text is generally well written and quite clear, I think that some improvement in the English presentation is still required. Despite the comments of one of the other Referees, I still identify this issue quite regularly across the text. To give an example: "little derivation" perhaps means "small deviation". Other unclear expressions or typos are also present and should be improved. The Supplementary Material text has more of these issues, e.g. in S10: "statistic angle"  "statistical angle", "statistics angle"  "statistical angle", "infeasible"  "unfeasible", "Because, the statistics" is a sentence that probably should attach to the previous one.

Reply: We would like to thank the reviewer for the kind suggestion. We have further improved the English presentation.

REVIEWERS' COMMENTS

Reviewer #3 (Remarks to the Author):

The authors have addressed my remarks. However, the reply to some of them has not been fully satisfactory and some of these questions are still open. Nevertheless, these were minor remarks and I am therefore satisfied with the current status of the manuscript, thus recommending its publication in the present form.

Draft Only